# Association between wasting and inadequate breastfeeding practices among infants under six months in SNNPR and Somali regions of Ethiopia: A multilevel cross-sectional study

Bethel Getachew[1]*, Yemane Berhane[2], Yadeta Dessie[3,4], Walelegn W. Yallew[5], Hanna Y. Berhane[1], Sunny S. Kim[6]

1 Nutrition and Behavioral Sciences Department, Addis Continental Institute of Public Health, Addis Ababa, Ethiopia, 2 Epidemiology and Biostatistics Department, Addis Continental Institute of Public Health, Addis Ababa, Ethiopia, 3 School of Public Health, College of Health and Medical Sciences, Haramaya University, Harar, Ethiopia, 4 African Population and Health Research Center, Nairobi, Kenya, 5 Global Health and Health Policy Department, Addis Continental Institute of Public Health, Addis Ababa, Ethiopia, 6 Nutrition, Diets, and Health Unit Department, International Food Policy Research Institute (IFPRI), Washington, DC, United State of America

* bethelgaciph@gmail.com

## Abstract

### Background

Wasting is a severe threat to children's survival and development. Attaining optimal breastfeeding practices for infants under six months of age remains a significant challenge in low-income countries. This study assessed the association between wasting and breastfeeding practices among infants under six months of age in the SNNPR and Somali regions of Ethiopia.

### Methods

The study used data from a large feasibility study conducted in the SNNPR and Somali regions of Ethiopia, from August-September 2021. This study involved 895 infants under six months of age with their mothers. The Poisson regression model with robust variance estimation was used to produce adjusted prevalence ratios (APRs) with 95% confidence intervals (CIs).

### Results

The prevalence of wasting was 16.5% (95% CI: 14.2, 19.2) among infants under six months of age. Non-exclusive breastfeeding (APR = 1.50; 95% CI:1.02, 2.21), delayed initiation of breastfeeding (APR = 1.52; CI:1.00, 2.30), being male infants (APR = 1.50; 95% CI:1.09, 2.07), and mothers who attained primary level (APR = 0.62; 95% CI: 0.40, 0.95) or secondary level education (APR = 0.30; 95% CI: 0.09, 0.99) were independently associated with wasting in the multivariable analysis.

**Data availability statement:** All relevant data are within the manuscript and its Supporting Information files.

**Funding:** The author(s) received no specific funding for this work.

**Competing interests:** The authors have declared that no competing interests exist.

## Conclusion

This study indicates a high prevalence of wasting among infants under six months of age. Non-exclusive breastfeeding and delayed initiation of breastfeeding were the modifiable factors significantly linked to infant wasting. Strengthening breastfeeding promotion and support may help reduce wasting in infants under six months.

## Background

Wasting is a significant public health challenge impacting infants below six months of age, particularly in low and middle-income countries [1]. Wasting among infants less than six month has serious consequences for morbidity and mortality [2]. The first six months of life are marked by rapid growth and neurological development, during which optimal nutrition, especially through exclusive breast feeding is essential [3]. This study aims to investigate the relationship between breastfeeding practice and wasting among infants under six months of age in Ethiopia, addressing a critical gap in understanding wasting in this vulnerable age group.

Globally, 48 percent of infants under six months of age are exclusively breastfed [4]. However, suboptimal breastfeeding practices, such as delayed initiation of breastfeeding, discarding colostrum, pre-lacteal feeding, and non-exclusive breastfeeding are recognized as contributors to undernutrition among infants [5]. These poor feeding practices can leave infants predisposed to sub-optimal growth and development [6]. Addressing suboptimal breastfeeding practices and promoting optimal nutrition is crucial in mitigating the risk of wasting among infants under six months of age [7].

Wasting is defined as having a weight-for-length z-score (WLZ) below −2 standard deviations from the median of the World Health Organization (WHO) child growth standards [8]. Wasting is an acute undernutrition that results from inadequate food consumption, usually from an incident of illness [9]. It is characterized by weight loss and a rapid worsening in nutritional status over a short period [10].

Despite the implementation of various strategies and programs, including the National Nutrition Program (NNP) [13], the prevalence of wasting among children under five remains high in Ethiopia [14]. According to the Ethiopian Demographic and Health Survey (EMDHS) 2019, 7% of children under five were wasted, with the prevalence of wasting in the Somali region was 21.4% and 6.3% in Southern Nations Nationalities and People's Region (SNNPR). Additionally, 9.4% of infants under six months of age were identified as wasted at the national level [11].

Previous studies have predominantly focused on children in the older age group, up to five years of age. This may be due to a poor understanding of the malnutrition burden in infants under six months of age [12] and challenges in accurately measuring the length of infants, especially in community settings, since small infants tend to be in knee-bent position which is a natural posture where the infant's legs are not fully straight but rather flexed at the knees. Moreover, calculating the WLZ using the WHO 2006 growth standards presents a challenge for infants with a length less than 45 cm, particularly for those born preterm or small-for-gestational-age.

Previous studies identified multiple risk factors associated with wasting among children up to five years of age, such as male sex, inadequate child feeding practices such as non-exclusive breastfeeding, late initiation of breastfeeding, pre-lacteal feeding, and early initiation of complementary feeding [13–15]. Maternal characteristics, included low level of education, lack of

occupation, fewer antenatal care (ANC) visits, and poor maternal health status [16]. Additionally, household conditions such as lack of latrine availability and the use of unprotected drinking water sources contribute to child wasting [17,18].

The aim of this work was to examine the association between breastfeeding practices and wasting in infants under six months of age in the SNNPR and Somali regions of Ethiopia. This analysis was conducted using the end-line household survey data from a maternal nutrition intervention study. We also provide recommendations for future policy and program directions, focusing on improving maternal and child health outcomes.

## Methods

### Study design and setting

Secondary data analysis was conducted using the end-line household survey data from a large feasibility and impact study of maternal nutrition interventions, which was designed as a cluster-randomized evaluation [19]. The study was conducted in the SNNPR and Somali regions of Ethiopia, with the end-line survey carried out from August 1 to September 30, 2021. The Somali region is located in the eastern part of Ethiopia, predominantly rural, and has a substantial pastoralist population, with livestock serving as the primary foundation of their livelihoods. The SNNPR region is primarily agrarian. The Federal Government of Ethiopia and regional health bureaus and the Alive & Thrive (A&T) selected seven woredas (districts; three in Somali and four in SNNPR regions) as program areas based on having adequate health facility access and infrastructure.

### Study sample

Study clusters were defined based on the health center catchment areas. In the Somali region, 10 health center clusters were included within the three study woredas, and 20 clusters were included within the four study woredas in the SNNPR region. Using stratified randomization of the total 30 clusters across both regions, 15 were allocated to the intervention group and 15 were allocated to the control group. The analysis for this paper used the sample in the control group only. The end-line household survey included 944 women with infants under six months of age in the control areas. Forty-two infants were not present at the time of the survey, and seven infants did not have a complete anthropometric measurement, so they were excluded. Consequently, for the current analysis, 895 infants under six months of age, residing with their mothers and having complete anthropometric measurements, were included in the study.

### Data collection procedures

Household and socio-demographic characteristics, as well as maternal and infant-related variables, were collected using structured questionnaires administered through computer-assisted personal interviews. Anthropometric measurements were conducted following standardized procedures outlined in the Food and Nutrition Technical Assistance (FANTA) Anthropometric Indicators Measurement Guide [20]. The weight of infants was measured using a portable Seca digital balance with a precision of 0.1 kg. The mother's weight was measured alone; then the mother held her infant, whose weight was obtained by subtracting the mother's weight using the tare function. The instrument was calibrated before each weighing. Mothers' height was measured upright using a portable wooden stadiometer, and infants' length measurements were taken in a recumbent position using a standard wooden measuring board, recorded to the nearest 0.1 cm, and measured twice for precision.

For the original study, missing or ambiguous data on birth dates were managed by verifying records, using parental recall based on significant events, and consulting healthcare workers to estimate infants' ages. When exact birth dates remained uncertain, ages were approximated to the nearest month to maintain consistency in the analysis. These methods minimized the impact of missing or ambiguous data.

## Ethical considerations

Ethical approval for the original study was obtained from the Ethical Review Board of the Addis Continental Institute of Public Health (Ref No. ACIPH/IRB/008/2021). Prior to data extraction, permission to use the data was sought and secured from the principal investigator at Addis Continental Institute of Public Health. The original study obtained written informed consent from all participants. For children under six months of age, additional consent was provided by their parents or guardians. To ensure participant confidentiality, all personally identifiable information was excluded from the dataset before analysis.

## Study variables and measurements

**Outcome variables.** The primary study outcome variable was wasting among infants under six months of age, defined as an infant having a weight-for-length (WLZ) z-score below −2 standard deviations from the median of WLZ, based on the WHO 2006 Child Growth Standards [21]. The outcome was constructed by using the zscore06 Stata command [22]. We marked anthropometric indices as outliers using the 2006 WHO recommendations if they were, > 5 or < −5, for WLZ [23].

**Explanatory variables.** Maternal variables included age, education level (no schooling, any primary school, and any secondary school or above); the number of antenatal care (ANC) visits (1−3 visits and four visits or above); place of delivery (home or health facility); and maternal height (short, medium, or tall).

The index child was the specific child selected for the study during data collection, chosen from among the children available in the household. Infant-level variables were categorized as follows: sex was coded as "male" and "female"; the age of the infant was recorded in completed months; and perceived birth size was coded as very small, average, or very big.

Infant feeding indicators were the proportion of infants aged 0–5 months who are fed exclusively on breast milk, with no other food or drink, including water, except for oral rehydration solutions (ORS) and drops or syrups containing vitamins, minerals, and medicine [24]. The following indicators were calculated based on the maternal report of infant feeding practices for the index child and list-based questions of all foods and liquids given to the infant in the previous 24h: infant ever breastfeeds; timing of breastfeeding initiation; colostrum feeding; pre-lacteal feeding; bottle feeding; started cow/goat/camel milk; started solid, semi-solid, and soft food were coded as "no" and "yes."

Exclusive breastfeeding was assessed based on the infant's feeding practices during the 24 hours preceding the survey. Infants who consumed no foods or liquids other than breast milk were coded as "exclusively breastfed," while others were coded as "not exclusively breastfed." Two key indicators (early initiation of breastfeeding and exclusive breastfeeding) were used for further analysis of associated factors.

Household characteristics were categorized as follows: sources of drinking water were coded as improved source (piped into dwelling, plot, yard), another improved source (protected well, spring, rainwater), or non-improved source (pond, lake, dam, river, open well, spring). Treatment of drinking water were coded as "no" and "yes". Toilet facilities were coded as follows; non-improved facility (pit latrine/traditional pit toilet), and improved facility (ventilated improved pit latrine (VIP) or no facility/bush/field).

The household wealth index was formulated by employing twenty-one variables related to the household's ownership of a selected set of assets [25]. Principal component analysis (PCA) was utilized to calculate the wealth index. These components were then categorized into five quintiles.

## Data analysis

Data analysis was performed using STATA version 17 statistical software. Descriptive statistics were used to examine the sample characteristics.

The prevalence of wasting was determined using proportions with 95% confidence intervals (CIs), to indicate the precision of the estimates. The association of independent variables with infant wasting was analyzed through Poisson regression with robust variance estimation, controlled for region (dummy variable) and geographical clustering.

All the variables with a p-value of less than 0.25 were used during initial screening in the bivariable analysis to capture variables that may not show strong individual associations with the outcome but could be influential when considered together with other predictors [26]. Variables with a p-value below 0.25 were included in the bivariable analysis to identify potential predictors that might have an influence when combined with others. For the multivariable analysis statistical significance was declared at a p-value < 0.05. Adjusted prevalence ratios (APRs) with 95% CI were utilized to present the results.

Four regression models were developed to assess the association of independent variables with infant wasting. Model I was adjusted for socio-demographic variables only, Model II was adjusted for maternal-related variables only, Model III was adjusted for infant-related variables only, and Model IV (combined model) was adjusted for socio-demographic, maternal-related, and infant-related variables; Table 5 shows which variables were included in each of the Models I-IV. To address multicollinearity, the variance inflation factor (VIF) was employed. The number of births was omitted from the final model due to multicollinearity with the number of pregnancies. The mean VIF for the combined model (Model IV) was 1.43, indicating the absence of multicollinearity among independent variables. Model fitness was assessed using the Hosmer–Lemeshow goodness-of-fit test. The Akaike information criterion (AIC) and the log-likelihood ratio test were used to estimate the goodness of fitness of the models. Based on the highest value of the Log-likelihood test and with the lowest values of AIC, the combined model (Model IV) was selected.

## Results

### Socio-demographic characteristics

The study included 895 infants under the age of six months and their mothers. The mean age of the mothers was 28.1 years. Twenty-eight percent of the mothers were aged between 25 and 29 years, and only 7.5% were aged between 15 and 19 years. More than half of the mothers (68.0%) were housewives, and 52.3% had no formal education. The majority of the study participants(67.2%) were from the SNNPR region, while the remainder (32.8%) were from the Somali region. Most of the respondents(63.1%) used improved sources of drinking water, while the rest (36.9%) used unimproved sources such as ponds, lakes, dams, rivers, open wells, and springs (Table 1).

### Maternal characteristics

Regarding maternal height, 38.7% of the mothers were categorized as short, 29.6% as medium, and 31.7% as tall. More than half of the respondents made four or more antenatal care (ANC) visits, while 42.1% had one to three ANC visits during their recent pregnancy.

**Table 1. Socio-demographic characteristics of the study participants among infants under six months of age in SNNPR and Somali Regions of Ethiopia, 2024 (n = 895).**

| Variable | Frequency | Percentage (%) |
|---|---|---|
| Age of the mother (years) | | |
| 15–19 | 67 | 7.5 |
| 20–24 | 173 | 19.3 |
| 25–29 | 253 | 28.3 |
| 30–34 | 219 | 24.5 |
| >35 | 183 | 20.4 |
| Mother education | | |
| No formal education | 468 | 52.3 |
| Primary education | 366 | 40.9 |
| Secondary and above | 61 | 6.8 |
| Father education | | |
| No formal education | 322 | 37.1 |
| Primary education | 421 | 48.6 |
| Secondary and above | 124 | 14.3 |
| Mother Occupation | | |
| Housewife | 609 | 68.0 |
| Other | 286 | 32.0 |
| Source of drinking water | | |
| Improved source (Piped into dwelling, plot, yard) | 381 | 42.6 |
| Other improved sources (Protected well, spring, rainwater) | 184 | 20.5 |
| Non-improved source (pond, lake, dam, river, open well, spring) | 330 | 36.9 |
| Treat drinking water | | |
| No | 760 | 84.9 |
| Yes | 135 | 15.1 |
| Toilet facility | | |
| Non-improved facility (Pit latrine/traditional pit toilet) | 599 | 66.9 |
| Improved facility (Ventilated improved pit latrine (VIP) | 35 | 3.9 |
| No facility/bush/field | 261 | 29.2 |
| Wealth index | | |
| Poor | 184 | 20.6 |
| Poorer | 174 | 19.4 |
| Middle | 179 | 20.0 |
| Rich | 179 | 20.0 |
| Richer | 179 | 20.0 |
| Region | | |
| SNNPR | 601 | 67.2 |
| Somali region | 294 | 32.8 |

Regarding the number of living children, 33.9% had more than five children living with them, while 33.5% had one to two living children and 38.6% of the mothers had more than five pregnancies. The majority of the respondents, 93.5%, gave birth to their most recent baby through a vaginal delivery. Sixty-eight percent of the index children were delivered at the health facility. Approximately 65.6% of the mothers received counseling on breastfeeding after delivery of the recent baby, while the remaining 34.4% did not receive counseling on breastfeeding (Table 2).

**Table 2. Maternal characteristics among mothers of infants under six months of age in SNNPR and Somali Regions of Ethiopia, 2024 (n = 895).**

| Variable | Frequency | Percentage (%) |
|---|---|---|
| Maternal Height (m) | | |
| Short | 342 | 38.7 |
| Medium | 261 | 29.6 |
| Tall | 280 | 31.7 |
| Number of pregnancies, Including the index child | | |
| 1–2 | 277 | 30.9 |
| 3–4 | 273 | 30.5 |
| ≥5 | 345 | 38.6 |
| Number of births, Including the index child | | |
| 1–2 | 291 | 32.5 |
| 3–4 | 275 | 30.7 |
| ≥5 | 329 | 36.8 |
| Number of living children, Including the index child | | |
| 1–2 | 300 | 33.5 |
| 3–4 | 292 | 32.6 |
| ≥5 | 303 | 33.9 |
| First ANC visit for the index child (month) | | |
| <3 month | 211 | 23.6 |
| 3–6 month | 546 | 61.0 |
| >6 month | 138 | 15.4 |
| Frequency of ANC visits for the index child | | |
| 1–3 | 377 | 42.1 |
| ≥4 | 518 | 57.9 |
| Place of delivery of the index child | | |
| Home | 284 | 31.7 |
| Health facility | 611 | 68.3 |
| Mode of delivery of the index child | | |
| Vaginal delivery | 837 | 93.5 |
| Caesarian section | 58 | 6.5 |
| Received breastfeeding counseling after the birth of the index child | | |
| No | 308 | 34.4 |
| Yes | 587 | 65.6 |

## Infant characteristics

Of the total infants included in the study, 49.3% were female, and 50.7% were male. The mean age of the infants was 2.8 ± 1.68 months. Only 12 infants (1.3%) were not breastfed. Among the breastfed infants, 88.5% initiated breastfeeding within the first hour of birth, and the majority, 90.8%, received colostrum. Pre-lacteal feeding was initiated for only 8.9% of the infants, and 72.2% were exclusively breastfed. (Table 3).

## Wasting among infants under six months of age

The overall prevalence of wasting among infants was 16.5% (95% CI: 14.1, 19.2). Within this group, male infants had a higher prevalence, accounting for 58.2% of wasted cases, while female infants accounted for 41.8%.

**Table 3. Infant under six months of age characteristics and related factors in SNNPR and Somali Regions of Ethiopia, 2024 (n = 895).**

| Variables | Frequency | Percentage (%) |
|---|---|---|
| Child sex | | |
| Male | 454 | 50.7 |
| Female | 441 | 49.3 |
| Child age (completed months) | | |
| 0 | 97 | 10.8 |
| 1 | 134 | 15.0 |
| 2 | 174 | 19.4 |
| 3 | 145 | 16.3 |
| 4 | 127 | 14.1 |
| 5 | 218 | 24.4 |
| Ever breastfeed | | |
| No | 12 | 1.3 |
| Yes | 883 | 98.7 |
| Timing of breastfeeding initiation | | |
| Within 1 hour | 782 | 88.6 |
| After 1 hour | 101 | 11.4 |
| Colostrum given | | |
| No | 81 | 9.2 |
| Yes | 802 | 90.8 |
| Pre-lacteal feeding, within the first two days | | |
| No | 804 | 91.1 |
| Yes | 79 | 8.9 |
| Bottle feeding | | |
| No | 627 | 70.1 |
| Yes | 268 | 29.9 |
| Exclusive breastfeeding | | |
| No | 249 | 27.8 |
| Yes | 646 | 72.2 |
| Frequency of breastfeeding, yesterday during the day and night | | |
| < 8 times | 177 | 19.8 |
| > 8 times | 702 | 78.4 |
| Not breastfeed | 16 | 1.8 |
| Started cow/goat/camel milk before six months | | |
| No | 719 | 80.3 |
| Yes | 176 | 19.7 |
| Started solid, semi-solid, and soft food before six months | | |
| No | 866 | 96.8 |
| Yes | 29 | 3.2 |
| Perceived birth size of the index child | | |
| Very small | 291 | 32.5 |
| Average | 421 | 47.0 |
| Very big | 183 | 20.5 |
| Wasted | | |
| Yes | 141 | 16.5 |
| No | 710 | 83.5 |
| Stunted | | |

*(Continued)*

**Table 3.** (Continued)

| Variables | Frequency | Percentage (%) |
|---|---|---|
| Yes | 86 | 10.0 |
| No | 773 | 90.0 |
| Underweight | | |
| Yes | 61 | 6.9 |
| No | 828 | 93.1 |

The age distribution of children in the wasted group was 11.4% aged zero months, 8.5% aged one month, 20.6% aged two months, and 59.5% aged three to five months. The age distribution of children in the non-wasted group was 9.3% aged zero months, 15.6% aged one month, 20.0% aged two months, and 55.1% aged three to five months. A high proportion of wasted children(66.7%) were from mothers with no formal education, indicating an association between lower maternal education and higher rates of child wasting. Among mothers with primary education, 31.2% of children were wasted, which is lower than those with no formal education. Only 2.1% of wasted children were from mothers with secondary or higher education.

Regarding antenatal care visits, 59.3% of infants whose mothers had four or more ANC visits were not wasted. For infants who were exclusively breastfed, 73.9% were not wasted. Additionally, 89.7% of infants who began breastfeeding within the first hour of birth were not wasted. 97.3% of infants were not wasted, among infants who did not start solid, semi-solid, and soft food before six months of age (Table 4).

## Factors associated with infant wasting

Infants born to mothers with primary education had a 38% lower prevalence of wasting compared to those born to mothers with no formal education (APR = 0.62; 95% CI: 0.40, 0.95). Infants born to mothers with secondary education had a 70% lower prevalence of wasting compared to those born to mothers with no formal education (APR = 0.30; 95% CI: 0.09, 0.99). Male infants had a 1.50 times higher prevalence of wasting than female infants (APR = 1.50; 95% CI:1.09, 2.07). Infants who were not exclusively breastfed were 1.50 times more likely to experience wasting compared to those who were exclusively breastfed (APR = 1.50; 95% CI: 1.02–2.21). Wasting among infants who initiated breastfeeding more than one hour after birth were 1.52 times more likely compared to infants who started breastfeeding less than one hour after birth (APR = 1.52; 95% CI: 1.00, 2.30). (Table 5).

## Discussion

This research aimed to assess the socio-demographic, maternal, and infant factors associated with wasting among infants under six months of age in the SNNPR and Somali regions of Ethiopia. The study identified significant associations between infant wasting and exclusive breastfeeding practice, initiation of breastfeeding within one hour of birth, male infants, and the mother's education level.

The prevalence of wasting in the current study was 16.5% (95% CI: 14.2, 19.2), indicating that wasting is a significant public health concern in the study area. The prevalence of wasting was consistent with previous research using data from the Ethiopian Demographic Health Survey of 2011 and 2016, which reported a prevalence of 14.2% [14]. Similarly, the finding was consistent with studies conducted in Bangladesh's Barisal district (18.8%) [27], Nairobi county (14.2%) [28], and Northern Tanzania (16.0%) [29]. However, research conducted in

**Table 4. Prevalence of different contributing factors for wasting among infants under six months of age in SNNPR and Somali Regions of Ethiopia, 2024.**

| Variables | Wasting | | Prevalence (95% CI) |
|---|---|---|---|
| | No, n (%) | Yes, n (%) | |
| Mother Age | | | |
| 15–19 | 55 (7.7) | 7 (4.9) | 11.2 (4.6, 21.8) |
| 20–24 | 145 (20.4) | 24 (17.0) | 14.2 (9.3, 20.3) |
| 25–29 | 193 (27.2) | 43 (30.5) | 18.2 (13.5, 23.7) |
| 30–34 | 170 (23.9) | 40 (28.4) | 19.0 (13.9, 25.0) |
| >35 | 147 (20.8) | 27 (19.2) | 15.5 (10, 21.7) |
| Mother Education | | | |
| No formal education | 348 (49.0) | 94 (66.7) | 21.2 (17.5, 25.3) |
| Primary education | 305 (43.0) | 44 (31.2) | 12.6 (9.3, 16.5) |
| Secondary and above education | 57 (8.0) | 3 (2.1) | 5.0 (1.0, 13.9) |
| Father Education | | | |
| No formal education | 243 (35.3) | 62 (45.9) | 20.3 (15.9, 25.2) |
| Primary education | 335 (48.6) | 65 (48.2) | 16.2 (12.7, 20.2) |
| Secondary and above education | 111 (16.1) | 8 (5.9) | 6.0 (2.9, 12.8) |
| Wealth Index | | | |
| Poor | 138 (19.5) | 34 (24.1) | 19.7 (14.0, 26.5) |
| Poorer | 132 (18.6) | 34 (24.1) | 20.4 (14.6, 27.4) |
| Middle | 145 (20.4) | 24 (17.0) | 14.2 (9.3, 20.3) |
| Rich | 145 (20.4) | 26 (18.5) | 15.2 (10.1, 21.4) |
| Richer | 150 (21.1) | 23 (16.3) | 13.2 (8.6, 19.2) |
| Number of pregnancies, Including the index child | | | |
| 1–2 | 225 (31.7) | 37 (26.3) | 14.1 (10.1, 18.9) |
| 3–4 | 218 (30.7) | 41 (29.0) | 15.8 (11.6, 20.8) |
| ≥5 | 267 (37.6) | 63 (44.7) | 19.0 (14.9, 23.7) |
| Place of delivery of the index child | | | |
| Home delivery | 214 (30.1) | 52 (36.9) | 19.5 (14.9, 24.8) |
| Health facility delivery | 496 (69.9) | 89 (63.1) | 15.2 (12.3, 18.3) |
| Number of ANC visits for the last pregnancy | | | |
| 1–3 | 289 (40.7) | 67 (47.5) | 18.8 (14.8, 23.2) |
| ≥4 | 421 (59.3) | 74 (52.5) | 14.9 (11.9, 18.4) |
| Received breastfeeding counseling after the birth of the index child | | | |
| No | 237 (33.4) | 56 (39.7) | 19.1 (14.7, 24.0) |
| Yes | 473 (66.6) | 85 (60.3) | 15.2 (12.3, 18.4) |
| Child sex | | | |
| Male | 346 (48.7) | 82 (58.2) | 19.1 (15.5, 23.2) |
| Female | 364 (51.3) | 59 (41.8) | 13.9 (10.7, 17.6) |
| Child Age (completed months) | | | |
| 0 | 66 (9.3) | 16 (11.4) | 19.5 (11.5, 29.7) |
| 1 | 111 (15.6) | 12 (8.5) | 9.7 (5.1, 16.4) |
| 2 | 142 (20.0) | 29 (20.6) | 16.9 (11.6, 23.4) |
| 3 | 114 (16.0) | 25 (17.7) | 17.9 (11.9, 25.3) |
| 4 | 102 (14.4) | 22 (15.6) | 17.7 (11.4, 25.6) |
| 5 | 175 (24.7) | 37 (26.2) | 17.4 (12.5, 23.2) |
| Exclusive breastfeeding | | | |
| No | 185 (26.1) | 53 (37.6) | 22.2 (17.1, 28.0) |
| Yes | 525 (73.9) | 88 (62.4) | 14.3 (11.6, 17.3) |

*(Continued)*

**Table 4.** (Continued)

| Variables | Wasting | | Prevalence (95% CI) |
|---|---|---|---|
| | No, n (%) | Yes, n (%) | |
| Timing of Breastfeeding initiation | | | |
| After 1 hour | 72 (10.3) | 21 (15.0) | 22.5 (14.5, 32.4) |
| Within 1 hour | 628 (89.7) | 118 (84.0) | 15.8 (13.2, 18.6) |
| Started solid, semi-solid, and soft food before six months | | | |
| No | 691 (97.3) | 133 (94.3) | 16.1 (13.6, 18.8) |
| Yes | 19 (2.7) | 8 (5.7) | 29.6 (13.7, 50.1) |
| Started cow/goat/camel milk before six months | | | |
| No | 577 (81.3) | 105 (74.5) | 15.3 (12.7, 18.3) |
| Yes | 133 (18.7) | 36 (25.5) | 21.3 (15.3, 28.2) |

India using the National Family Health Survey-3 data reported a higher prevalence of wasting, 31.0% [30]. This higher prevalence of wasting may be due to low exclusive breastfeeding rates during the first six months of life, with only 18.6% exclusively breastfeeding between 4 and 6 months, compared to this study, where 72.2% of infants were found to be exclusively breastfed. Likewise, a study conducted in urban areas of Indonesia showed a high prevalence of wasting, 25.6% [31]. This may be due to the study's findings indicating a high prevalence of early initiation of complementary feeding before six months of age, which could contribute to various infectious diseases and undernutrition. The prevalence of wasting in the current study (16.5%) is higher than a study conducted in Deder Woreda, East Hararge Zone, and Jimma Zone, Ethiopia, where the prevalence was 13.3% [12]. The lower prevalence of wasting may be due to the difference in the sample size and the geographical area. The findings of the study collectively emphasize the challenge of wasting across different geographical regions, highlighting the need for targeted interventions to address the issue of wasting.

Educated mothers were less likely to have wasted infants than mothers without formal education. This result aligns with studies conducted in Debre Berhan town, Ethiopia [16], Bangladesh Barisal district [27], and a study carried out in Pakistan among children under two years of age [32]. The mother's educational level can significantly impact various aspects of infant health, including the risk of infant wasting [33]. Educated mothers are more likely to understand the importance of proper nutrition during pregnancy and infancy [34]. Moreover, they may also have a better understanding of breastfeeding practices, infants' nutrition, the importance of clean water, proper sanitation, and hygiene in preventing diseases that can contribute to undernutrition [35].

Infants under six months who were not exclusively breastfed had a 50% higher prevalence of wasting. This finding is consistent with a study by Nigatu et al., which analyzed data from the Ethiopian Demographic and Health Surveys (2011–2016) and reported a 45% increase in the odds of wasting among infants who were not exclusively breastfed [14]. Similarly, a study conducted in Bangladesh found that the lack of exclusive breastfeeding increased the odds of wasting by 79% [36]. A study conducted in Nigeria also reported a significant positive association between the absence of exclusive breastfeeding and wasting [37]. According to the 2019 Ethiopia Demographic and Health Survey, the prevalence of exclusive breastfeeding for infants under six months in the SNNPR was 58%. In contrast, the rate in the Somali Region was lower, 25% [11]. Breastfeeding is a widely recognized option for child feeding that provides immediate and long-term protection against infections. This protection is attributed to its abundant supply of immune factors, anti-microbial, and anti-inflammatory agents [38]. Moreover, introducing any drink or food other than breast milk, particularly before the age

**Table 5. Factors associated with wasting among infants under six months of age in SNNPR and Somali Regions of Ethiopia, 2024.**

| Variables | CPR (95% CI) | Model I (Socio-demographic factors) APR (95% CI) | Model II (maternal related factors) APR (95% CI) | Model III (infant related factors) APR (95% CI) | Final model APR (95% CI) |
|---|---|---|---|---|---|
| Mother Age | | | | | |
| 15–19 | 1 | 1 | | | 1 |
| 20–24 | 1.25 (0.57, 2.77) | 1.53 (0.66, 3.50) | | | 1.45 (0.63, 3.32) |
| 25–29 | 1.61 (0.76, 3.41) | 1.90 (0.84, 4.25) | | | 2.03 (0.89, 4.63) |
| 30–34 | 1.68 (0.79, 3.57) | 1.68 (0.73, 3.82) | | | 1.62 (0.66, 3.93) |
| >35 | 1.37 (0.63, 2.99) | 1.25 (0.53, 2.94) | | | 1.11 (0.43, 2.87) |
| Mother Education | | | | | |
| No formal education | 1 | 1 | | | 1 |
| Primary education | 0.59 (0.42, 0.82) | 0.59 (0.39, 0.89)* | | | 0.62 (0.40, 0.95)* |
| Secondary and above education | 0.23 (0.07, 0.71) | 0.28 (0.09, 0.89)* | | | 0.30 (0.09, 0.99)* |
| Father Education | | | | | |
| No formal education | 1 | 1 | | | 1 |
| Primary education | 0.79 (0.58, 1.09) | 1.02 (0.70, 1.47) | | | 1.06 (0.72, 1.56) |
| Secondary and above education | 0.33 (0.16, 0.66) | 0.50 (0.22, 1.10) | | | 0.50 (0.22, 1.14) |
| Wealth Index | | | | | |
| Poor | 1.41 (0.85, 2.31) | 0.93 (0.53, 1.62) | | | 0.81 (0.45, 1.46) |
| Poorer | 1.68 (1.04, 2.69) | 1.10 (0.66, 1.85) | | | 1.12 (0.66, 1.88) |
| Middle | 1.04 (0.60, 1.78) | 0.73 (0.41, 1.29) | | | 0.73 (0.42, 1.29) |
| Rich | 1.17 (0.69, 1.96) | 0.97 (0.56, 1.66) | | | 0.99 (0.58, 1.72) |
| Richer | 1 | 1 | | | 1 |
| Number of pregnancies, Including the index child | | | | | |
| 1–2 | 1 | | 1 | | 1 |
| 3–4 | 1.12 (0.74, 1.68) | | 1.12 (0.74, 1.69) | | 1.02 (0.64, 1.62) |
| ≥5 | 1.35 (0.93, 1.96) | | 1.33 (0.91, 1.93) | | 1.17 (0.70, 1.95) |
| Place of delivery of the index child | | | | | |
| Home delivery | 1.28 (0.94, 1.75) | | 1.12 (0.78, 1.62) | | 0.83 (0.55, 1.27) |
| Health facility delivery | 1 | | 1 | | 1 |
| Number of ANC visits for the last pregnancy | | | | | |
| 1–3 | 1 | | 1 | | 1 |
| ≥4 | 0.79 (0.58, 1.07) | | 0.84 (0.60, 1.17) | | 0.85 (0.60, 1.19) |
| Received breastfeeding counseling after the birth of the index child | | | | | |
| No | 1.25 (0.92, 1.70) | | 1.14 (0.81, 1.60) | | 1.14 (0.81, 1.59) |
| Yes | 1 | | 1 | | 1 |
| Child sex | | | | | |
| Male | 1.37 (1.01, 1.86) | | | 1.40 (1.02, 1.92)* | 1.50 (1.09, 2.07)* |
| Female | 1 | | | 1 | **1** |
| Child Age (completed months) | | | | | |
| 0 | 1.11 (0.65, 1.89) | | | 1.38 (0.80, 2.38) | 1.31 (0.74, 2.33) |
| 1 | 0.55 (0.30, 1.03) | | | 0.62 (0.32, 1.20) | 0.62 (0.31, 1.24) |
| 2 | 0.97 (0.62, 1.51) | | | 1.18 (0.74, 1.87) | 1.19 (0.74, 1.91) |
| 3 | 1.03 (0.65, 1.63) | | | 1.24 (0.77, 1.99) | 1.27 (0.80, 2.03) |
| 4 | 1.01 (0.62, 1.64) | | | 1.10 (0.68, 1.78) | 1.13 (0.70, 1.83) |
| 5 | 1 | | | 1 | 1 |

*(Continued)*

**Table 5.** (Continued)

| Variables | CPR (95% CI) | Model I (Socio-demographic factors) APR (95% CI) | Model II (maternal related factors) APR (95% CI) | Model III (infant related factors) APR (95% CI) | Final model APR (95% CI) |
|---|---|---|---|---|---|
| Exclusive breastfeeding | | | | | |
| No | 1.55 (1.14, 2.10) | | | 1.48 (1.03, 2.13)* | 1.50 (1.02, 2.21)* |
| Yes | 1 | | | 1 | 1 |
| Timing of Breastfeeding initiation | | | | | |
| Within 1 hour | 1 | | | 1 | 1 |
| After 1 hour | 1.42 (0.94, 2.15) | | | 1.46 (0.96, 2.22) | 1.52 (1.00, 2.30)* |
| Started solid, semi-solid, and soft food before six months | | | | | |
| No | 0.54 (0.29, 0.99) | | | 0.63 (0.33, 1.19) | 0.62 (0.31, 1.23) |
| Yes | 1 | | | 1 | 1 |
| Started cow/goat/camel milk before six months | | | | | |
| No | 0.72 (0.51, 1.01) | | | 1.05 (0.69, 1.59) | 1.10 (0.71, 1.71) |
| Yes | 1 | | | 1 | 1 |
| Model fit statistics | | | | | |
| Log-likelihood | | −366.10 | −391.70 | −380.03 | −350.63 |
| AIC | | 758.2 | 795.4 | 782.06 | 757.26 |

p < 0.25 in crude prevalence ratio (CPR); * p < 0.05 in adjusted prevalence ratio (APR).

of four months, is associated with an increased risk of gastrointestinal diseases. This increased risk may lead to growth retardation, micronutrient deficiencies, and heightened vulnerability to various infectious diseases and undernutrition [39].

Our study's findings underscore that exclusive breastfeeding is far more than a feeding choice; it is a crucial, evidence-based intervention. These results add to the growing body of evidence affirming the essential role of exclusive breastfeeding in promoting healthy growth and reducing the risk of wasting in infants during their early months. Strengthening efforts to support exclusive breastfeeding is therefore vital in combating infant malnutrition. This reinforces the importance of breastfeeding as a powerful intervention for infant health, and mothers should be encouraged and well-supported to exclusively breastfeed as a core strategy for improving child health outcomes and reducing malnutrition.

This study found that infants who did not initiate breastfeeding within one hour of birth had 52% higher odds of wasting compared to those who started breastfeeding within the first hour. This finding aligns with research conducted across 20 developing countries using demographic health survey data, which reported a 31% increase in the odds of wasting among infants who delayed breastfeeding initiation [40]. Similarly, a study from South Asia showed that early breastfeeding initiation reduced the odds of wasting by 8% [41]. These findings highlight the importance of timely breastfeeding initiation in reducing the risk of wasting across various populations.

This could be due to early initiation of breastfeeding having a protective effect against infections and reducing newborn mortality. Evidence shows that the risk of morbidity due to diarrhea and other infections is significantly higher among newborns who experience late initiation of breastfeeding either partially or not breastfed at all [42]. Additionally, early breastfeeding initiation creates a positive foundation for exclusive breastfeeding by establishing an adequate milk supply, ensuring effective latching, providing nutrient-rich colostrum, and enhancing maternal-infant bonding [43]. This "critical window" within the first hour

after birth is a crucial period when infants are most alert and ready to begin breastfeeding. Natural behaviors such as rooting (the reflex to turn toward the breast) and suckling (the ability to latch and draw milk) are activated at birth, fostering a strong mother-infant bond and supporting successful early feeding [44]. However, an infant's readiness to breastfeed can be influenced by prenatal factors. Preterm infants or those with complications often have underdeveloped reflexes and slower physiological responses, which can hinder effective latching and feeding. Additionally, conditions like low birth weight or birth asphyxia can reduce alertness and weaken reflexes, leading to delays in breastfeeding initiation [45]. Furthermore, cultural beliefs and health misinformation create additional barriers, particularly in regions where traditional practices influence infant care. In some cultures, colostrum is viewed as "dirty" or "unhealthy," and caregivers may discard it, delaying the initiation of breastfeeding until what they perceive as the "clean milk" appears, typically a few days post-birth [46]. Moreover, some mothers are advised to delay feeding to let the infant "rest" or because of concerns about pain or discomfort associated with breastfeeding. These cultural barriers are often compounded by healthcare misinformation and inadequate counseling during prenatal care. Without proper guidance on the benefits of early breastfeeding, mothers may delay initiation or avoid it altogether [44]. Thus, promoting early breastfeeding initiation requires both infant readiness and robust education and awareness to counteract these cultural beliefs and misinformation.

This study found that males were 50% more likely to be wasted than females. This finding aligns with data from the 2019 Ethiopian Mini Demographic and Health Survey, which reported that being female was associated with 30% lower odds of wasting [15]. Similarly, a study conducted in three disadvantaged East African Districts, Rwanda, Uganda, and Tanzania among children under five years of age found that being female reduced the odds of wasting by 14% [47]. Furthermore, several studies have found that male children are more likely to experience wasting compared to female children[29,48–50]. Different literature emphasizes the importance of approaching the issue of undernutrition comprehensively, considering a variety of factors that can impact the nutritional status of infants [49,51]. Several factors potentially contribute to the nutritional disparities between boys and girls, including biological and hormonal differences [52,53], which have a significant impact on energy consumption, nutritional requirements, and vulnerability to diseases [54,55]. Additionally, an infant's nutritional well-being is collectively influenced by various factors such as poor maternal health [56,57], poor household conditions [58], and suboptimal breastfeeding practices [59]. Moreover, the higher vulnerability of male infants to wasting is explained by the gender allocation of resources in many societies, particularly in feeding practices, caregiving, and healthcare access. In some cultural contexts, boys and girls receive different levels of care based on societal preferences for one gender over the other. For instance, some cultures prioritize feeding male infants larger portions or more frequent feedings, which are thought to support their growth and future roles as providers or leaders [60]. In contrast, in cultures where girls are seen as more valuable for future reproductive roles or caregiving responsibilities, they may receive better nutrition during early childhood, which could shield them from wasting. In contrast, boys may face neglect or less attention, which can leave them more vulnerable to malnutrition and wasting [55].

Understanding these culturally rooted disparities is critical, as they suggest the need for research on how early caregiving practices and societal gender roles uniquely affect the growth and health of male and female infants. Such research would allow for targeted interventions addressing sex-specific vulnerabilities, ensuring that both boys and girls receive adequate care and nutritional support during the formative first six months of life [54]. Strengthening evidence-based policies to bridge these gender disparities in infant nutrition and health access is vital for reducing early-life malnutrition and promoting equal health opportunities for all infants.

While the wealth index showed a significant association with infant wasting in the bivariate analysis, this association did not hold in the multivariable analysis when other factors such as breastfeeding practices and maternal education, were considered. One possible explanation is that wealth may indirectly influence wasting by affecting more proximal factors, such as the mother's education level and access to health information, which in turn influence infant feeding practices [61]. When these factors are controlled for in the multivariable model, the direct association between wealth and wasting diminishes. Additionally, wealth may capture general household resources but may not fully reflect specific nutritional or healthcare practices that are crucial for preventing infant wasting. Thus, while wealthier households might have better resources, the actual practices around infant feeding and care are key determinants of infant nutrition are more directly relevant to wasting outcomes [62].

This analysis also explored whether the number of pregnancies and births (parity) was associated with infant wasting. Although higher parity is often thought to affect child health outcomes due to resource allocation or maternal experience, our findings did not show a significant association between the number of pregnancies, parity, and wasting in the multivariable analysis. One possible reason for this non-significance is that factors more directly tied to the immediate care and feeding of the infant, such as exclusive breastfeeding and early initiation of breastfeeding, may be stronger determinants of wasting. The mother's experience with multiple births may increase her general caregiving knowledge but does not necessarily ensure the specific practices that prevent wasting [63].

## Strengths of the study

The study used a comprehensive approach and robust dataset, which include a large sample of infants, enhancing the reliability and generalizability of its findings on infant wasting. By examining a variety of socio-demographic, maternal, and infant-related factors, such as maternal education, breastfeeding practices, and socio-economic status. Additionally, the study provides valuable insights into modifiable determinants of malnutrition and potential intervention points. Focusing on early-life factors, particularly feeding practices like exclusive breastfeeding, emphasizes a crucial developmental period during which timely interventions can significantly impact infant health outcomes, contributing to the prevention of wasting and other forms of malnutrition.

## Limitations of the study

A limitation of this study could be the cross-sectional nature of the design, which poses difficulty in examining the causal relationship between wasting status and independent variables. Additionally, there is potential recall bias that may arise when reporting on the infant's feeding practices. To minimize this recall bias, we included only the mothers of the infants and used a short recall period of 24 hours for certain feeding-related questions. Furthermore, while infant morbidity could be a potential determinant of wasting, no data on infant illness were collected. Similarly, information about whether the infants were singletons or part of multiple births was not collected.

## Conclusion

A high prevalence of wasting (16.5%) among infants under six months of age was observed in our study areas in the SNNPR and Somali regions of Ethiopia. Not breastfeeding exclusively, late initiation of breastfeeding, and low maternal education were significantly associated with infant wasting. The study's findings highlight the need to strengthen the national strategy for infant and young child feeding programs by promoting and creating awareness about

exclusive breastfeeding practices and early initiation of breastfeeding. Additionally, the study's findings emphasize the need to strengthen policies aimed at improving women's education to reduce the illiteracy rate among mothers.

## Supporting information

**S1 Dataset. The data set used to produce the study's findings.**
(XLS)

**S1 Table. Explanatory variables categorization and coding.**
(DOCX)

**S2 Table. Variables used for constructing the wealth index.**
(DOCX)

**S1 Fig. WHZ Z-score with Standard Normal Distribution.**
(TIF)

## Acknowledgements

We would like to express our gratitude to the Haramaya University School of Public Health for academic support to the first author. We acknowledge the Addis Continental Institute of Public Health for permitting us to utilize the data.

## Author contributions

**Conceptualization:** Bethel Getachew, Yemane Berhane, Yadeta Dessie, Walelegn W. Yallew, Hanna Y. Berhane, Sunny S. Kim.

**Data curation:** Bethel Getachew, Yemane Berhane, Hanna Y. Berhane, Sunny S. Kim.

**Formal analysis:** Bethel Getachew, Yemane Berhane, Yadeta Dessie, Walelegn W. Yallew, Hanna Y. Berhane, Sunny S. Kim.

**Funding acquisition:** Bethel Getachew, Yemane Berhane.

**Investigation:** Bethel Getachew, Yemane Berhane, Yadeta Dessie, Walelegn W. Yallew, Hanna Y. Berhane, Sunny S. Kim.

**Methodology:** Bethel Getachew, Yemane Berhane, Yadeta Dessie, Walelegn W. Yallew, Hanna Y. Berhane, Sunny S. Kim.

**Project administration:** Bethel Getachew, Yemane Berhane, Hanna Y. Berhane, Sunny S. Kim.

**Resources:** Bethel Getachew, Yemane Berhane, Hanna Y. Berhane, Sunny S. Kim.

**Software:** Bethel Getachew, Yemane Berhane, Hanna Y. Berhane, Sunny S. Kim.

**Supervision:** Bethel Getachew, Yemane Berhane, Yadeta Dessie, Walelegn W. Yallew, Hanna Y. Berhane, Sunny S. Kim.

**Validation:** Bethel Getachew, Yemane Berhane, Yadeta Dessie, Walelegn W. Yallew, Hanna Y. Berhane, Sunny S. Kim.

**Visualization:** Bethel Getachew, Yemane Berhane, Yadeta Dessie, Walelegn W. Yallew, Hanna Y. Berhane, Sunny S. Kim.

**Writing – original draft:** Bethel Getachew, Yemane Berhane, Yadeta Dessie, Walelegn W. Yallew.

**Writing – review & editing:** Bethel Getachew, Yemane Berhane, Yadeta Dessie, Walelegn W. Yallew, Hanna Y. Berhane, Sunny S. Kim.

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
