## [Decision Letter · Decision Letter 0]

1 Oct 2024

PONE-D-24-29945Association Between Wasting and Inadequate Breastfeeding Practices Among Infants Under Six Months in Southeast Ethiopia: A Multilevel Cross-Sectional StudyPLOS ONE

Dear Dr. Getachew,

Thank you for submitting your manuscript to PLOS ONE. After careful consideration, we feel that it has merit but does not fully meet PLOS ONE’s publication criteria as it currently stands. Therefore, we invite you to submit a revised version of the manuscript that addresses the points raised during the review process.

Take seriously the following major concerns regarding the data periods claimed. The dates of birth in the dataset are all in 2013 but the study claims to have been carried out in 2021. Justify this discrepancy.Along with global statistics, mention the statistics of exclusively breastfeeding in the study area.From the Study design and setting, delete the sentence “The study aimed to strengthen nutrition interventions integrated into existing antenatal care services and to improve maternal nutrition practices.The authors write “All the variables with a p-value of less than 0.25 in bivariable analysis were used for multivariable analysis.” What is the reason for using a p-value of less than 0.25? Justify your choice and add a reference(s). Is it “multivariable” or “multivariable” check it carefully. There is a huge difference between them. It is recommended to show the p-values in bivariate analysis.Limitations of the study should be a separate section.

We look forward to receiving your revised manuscript.

Kind regards,

Md. Moyazzem Hossain

Academic Editor

PLOS ONE

Journal Requirements: When submitting your revision, we need you to address these additional requirements. 1. Please ensure that your manuscript meets PLOS ONE's style requirements, including those for file naming. The PLOS ONE style templates can be found at https://journals.plos.org/plosone/s/file?id=wjVg/PLOSOne_formatting_sample_main_body.pdf and https://journals.plos.org/plosone/s/file?id=ba62/PLOSOne_formatting_sample_title_authors_affiliations.pdf 2. We suggest you thoroughly copyedit your manuscript for language usage, spelling, and grammar. If you do not know anyone who can help you do this, you may wish to consider employing a professional scientific editing service.  The American Journal Experts (AJE) (https://www.aje.com/) is one such service that has extensive experience helping authors meet PLOS guidelines and can provide language editing, translation, manuscript formatting, and figure formatting to ensure your manuscript meets our submission guidelines. Please note that having the manuscript copyedited by AJE or any other editing services does not guarantee selection for peer review or acceptance for publication.  Upon resubmission, please provide the following: The name of the colleague or the details of the professional service that edited your manuscript A copy of your manuscript showing your changes by either highlighting them or using track changes (uploaded as a *supporting information* file) A clean copy of the edited manuscript (uploaded as the new *manuscript* file)”

Reviewers' comments:

Reviewer's Responses to Questions

**Comments to the Author**

1. Is the manuscript technically sound, and do the data support the conclusions?

Reviewer #1: Partly

Reviewer #2: Yes

2. Has the statistical analysis been performed appropriately and rigorously? 

Reviewer #1: Yes

Reviewer #2: I Don't Know

3. Have the authors made all data underlying the findings in their manuscript fully available?

Reviewer #1: Yes

Reviewer #2: Yes

4. Is the manuscript presented in an intelligible fashion and written in standard English?

Reviewer #1: Yes

Reviewer #2: No

5. Review Comments to the Author

Reviewer #1: Thank you for the opportunity to review your work. The underlying approach is sound and many of the comments below are about making the work easier to read, or more precise. However, there is a major concern re. data periods claimed. The dates of birth in the dataset are all in 2013 but the study claims to have been carried out in 2021. It is for this reason I have recommended major revision.

MAJOR POINTS

INTRO

Authors take a long time to state this aim of this study. Suggest adding one or two lines to to the end of the first paragraph that states the aim of this work was to look at the association between breastfeeding practices and wasting in infants aged under six months in the South of Ethiopia.

METHODS

Data claims to be from 2021 but the included dataset has all of the children's date of birth in 2013.

RESULTS

Authors discuss LAZ and WAZ in Intro/Results but never refer to it in the manuscript again. Remove any mention of these variables if they are not discussed in detail outside of Table 3.

DISCUSSION

There is no discussion of null or negative findings. For example, a wealth index is included but is not addressed in the discussion. Similarly, the number of pregnancies and births should be discussed at least briefly as to why they do or not affect the results.

MINOR POINTS

Abstract

Line 26 - Suggest removing "inadequate".

Line 27 - South East, not Southeast.

Line 27 - SNNPR is in the South West of the country. Describing the work as being set only in South East is incorrect.

Introduction

Line 75 - Capitalisation of World Health Organisation

Lines 80-84 - This paragraph does not add to the value of the introduction. Consider removing it because the introduction is very long already.

Line 87 - "The prevalence of wasting..." should not be a new sentence but should carry on from the first sentence in the paragraph.

Line 92 - This paragraph starts by mentioned underlying factors but goes on to discuss problems with measurement methodologies. Consider revising first sentence. Line 102 onwards seems to discuss underlying factors but appears to be a new paragraph.

Line 99 - WLZ initialism was defined earlier but "Weight-for-Length Z-score" is written out in full here.

Methods

Line 124 and elsewhere - It is unclear what the "end-line" survey is.

Line 130 - Data period claims to be 2021 but dates of births in dataset are 2013.

Line 149 - Were all children singleton births? Multiple births can have lower birth weight which may be relevant to growth comparisons in the early months of life.

Line 154 - It is unclear what is meant by "enumerator training and field practice".

Line 171 - Is zcore06 function from a third-party package? If so, please include a citation.

Line 175 - LAZ and WAZ do not appear to be mentioned anywhere else in the article to please remove this line.

Line 178 - Was mother height (included in the supplied dataset) used in any of the models? If so, was it the continuous variable or the categorical in the dataset?

Line 183 - Was the "child perceived birth size" variable used anywhere? If so, this should be recognised as a limitation since it is subject to recall bias and a subjective variable.

Line 191 - What is an "index child"? No definition given.

Line 192 - Use semi-colons to delineate between items in the list.

Line 210 - The citation for the wealth index is a technical report but I couldn't find where it lists the consumer items. A more in depth explanation of how the wealth index is calculated in this study is required. This may be a supplemental file that lists the "20 items" that the authors describe, and how they used these to divide in to quintiles.

Line 237 - Suggest adding "only", e.g. "Model I was adjusted for 238 socio-demographic variables ONLY, Model II was adjusted for maternal-related variables ONLY..."

Methods general note - Explain that there may be some uncertainty around the child's day of birth so some ages were given to the nearest month, only. Explain how missing or ambiguous data were handled.

Line 241 - Please add a statement that Table 5 shows which variables were included in each of Models I-IV.

Line 241 - P values should not be conflated with confidence intervals. P values should not be mentioned here.

Results

Line 274 - ResultS rather than Result

Line 276 - Inconsistent use of numbers and proportions across this paragraph. Sometimes numbers and percentages are presented, sometimes just percentages.

Line 276 - Most of these statisticss are presented in Table 1. This paragaph could be much shorter.

Line 280 - SNNPR should be redefined in Methods section or use "Southern". Just be consistent in which term you use.

Line 283 - "Treat drinking water" - Treatment of drinking water is not mentioned in the variable definitions in the Methods section.

Line 293 - "Regarding the number of living children, 33.9% had more 294 than five children living with them, while 33.5% had one to two living children 295 and 38.6% of the mothers had more than five pregnancies." I found this sentence very difficult to follow. It made me wonder what is the importance of the number of pregnancies versus births, which is not discussed anywhere. Were pregnancies versus births a self-reported number, for example?

Line 305 - Sex percentages are different to the values presented in Table 3.

Line 321 - Why are there square brackets?

Line 322 - The prevalences (58 and 41.8) are much higher than the previous line (16.5%). Rewrite to clarify. Suggest that the prevalence is given for each sex, rather than the split by sex in the group of children in those that are wasted.

Line 323 - Suggest to describe trends with age in the wasted and non-wasted groups rather than listing specific values. The values are all in Table 4 for the interested reader.

Line 326 - Similarly, make a short description of any trend with wasting and education level but let the reader look at Table 4 for exact figures.

Line 340-343 - Repetitious of methods section (Lines 245-247). Remove here or remove from methods section.

Line 346 - P values described in Methods section but CI's are reported in results. Clarity should be made in Methods section to talk about CI's and what percentage CI is being used. Please also state 95% CI rather than just CI in the brackets.

Line 344 - I do not believe that describing an APR of 0.62 as 38% lower is technically correct, but this may just be a semantics issue.

Line 349 and 352 - Inconsistent levels of precision on the estimates. The estimate is rounded to 1.5 but CI given as 1.09, 2.07 for example. Stick to same number of sig fig (so report as 1.50 inc. trailing zero).

Line 352 - Suggest change to "Infants WHOM initiated breastfeeding MORE than one after birth..." to make it clear that these were children that experienced delayed first feeding.

Table 5 - Check levels of precision in all reported numbers. There are a couple of cases where a trailing zero is missing. For example, maternal age 25-29 column Model I = 1.9 (0.84-4.25) should be 1.90 and Pregnancies > 5 Final Model is 1.17(0.7-1.95) when it should be 1.17 (0.70-1.95).

Table 5 - CIs are reported with inconsistent formatting. Commas are used in the text but hypens in the Table. Please be consistent and use commas or hyphens throughout.

Table 5 - Bold font OR asterisks should be used to indicate sig values, not both.

Table 5 - Footer - I don't know what the note about crude prevalence refers to in this table, specifically.

Results general note - Did wealth index matter in terms of wasting prevalence?

Discussion

No comparison of the results with the most relevant study (citation #15, mentioned in introduction).

Line 372 - Unnecessary square brackets about prevalence estimate.

Line 377 - Nairobi COUNTY rather than country?

Lines 405-409 - Please report the prevalence ratios from the studies cited. This will allow the reader to see how these ratios compare in other contexts.

Line 417 - Please add something to the end of this paragraph that essentially says "Our results therefore add to the evidence that breastfeeding is a vitally effective practice in the first months of a child's life and mothers should be encouraged and supported.." etc etc

Line 430 - Please add further critical discussion about what feeding within the first hour means. Does it relate to the child's in utero development such that a better-developed newborn is more able to initiate feeding? Or is it a cultural or health misinformation issue (in the same way colostrum discarding is)? The authors have explained the associations and consequences of early feeding well, but more detail about why an infant may or may not feed in the first hour is important to this discussion point.

Line 447 - It seems important to mention there may be cultural factors underpinning this sex difference. Or at least for the authors to mention if there is a lack of research on potential cultural factors. Pre-pubescent children are relatively similar between sexes.

Line 453 - What would the implications be on your estimates of prevalence if you could account for morbidity? (Presumably children that experienced extreme wasting sadly died so would not be included, thereby making your estimate an underestimate?)

Line 454 - Authors should also list strengths of their study.

Line 457 - ...Southern regions OF ETHIOPIA.

Other points

Line 465 - AcknowledgmentS not Acknowledgment

References

URLs or DOIs should be added to the following citations: 1, 3, 4, 5, 8, 13, 15, 23, 25, 27, 34, 38

Reviewer #2: It was a pleasure to review a paper on such an important global issue. I believe that this article is technically sound, and that the data provided supports the conclusions drawn by the researchers. From my experience with statistical analysis, I believe that the analytical protocols were relevant, rigorous, and justified in context. However, my skills in this area are not to a level where I can confidently endorse the statistical analysis here. This does not at all mean that the analysis was not rigorous and appropriate, as per the publication guidelines, but simply that I am not in a position to make this judgment. I believe the authors have made available all relevant and supporting data, including the raw endline survey data as an appendix and all tables of summarized data.

For point 4 I had selected 'no' because I have a few suggestions here. Overall, this paper was written clearly and in correct English. But in some places, I think some rewording would improve clarity:

Line 97: What is meant by the 'knee-bent' position?

Line 99-101: The sentence beginning 'Moreover' could be rewritten, as it seems to change direction halfway through and is therefore a little unclear.

Line 105-107: The sentence beginning 'Maternal' - should 'including' be 'included'?

Line 187-189: Were these two sentences supposed to be just one sentence?

Lines 308 + 311: You have repeated data here.

Line 422-424: This sentence could be reworded to improve clarity. Maybe, "This could be due to early initiation of breastfeeding have a protective effect against infections and reducing newborn mortality."?

This note might be a little pedantic on my part, but I think replacing 'done' with 'completed' or 'conducted' reads better. (Lines 28, 112, and 422).

In the results section on wasting under 6 months (beginning Line 321) the percentages presented switch between representing the proportion of wasting and the proportion without wasting. E.g., "Among mothers with no formal education, 66.7% of their infants were wasted" (Line 326) and "Regarding antenatal visits, 59.3% of infants who has four or more ANC visits were not wasted" (Line 329). I think it would be clearer to consistently report either the percentage of wasting or percentage without wasting. This might also help to avoid readers misinterpreting the data here.

Three other thoughts I had whilst reviewing this paper:

1. In the introduction (Line 113) it states that the policy and program implications of the findings will be discussed. However, these are not discussed in the paper, but rather mentioned in the conclusion as a future consideration. In my view, a 'discussion' of the policy and program implications would include some detail on what is currently in place followed by suggestions (based on the findings) of how these could be updated/improved to reduce wasting in infants in this area. I would recommend including some discussion or rewording the introduction to change "We also discuss" to "We also make recommendations for future policy and program directions" (or something along those lines).

2. The survey options for number of ANC visits were "1-3 visits" or "4 or more visits". Upon review of the feasibility study from which this survey was taken, at least 1 ANC visit was an eligibility requirement for inclusion into this survey. Is it possible that there are women who have not attended any ANC visits in this area and were therefore not included in the survey? If so, this would be a limitation worth mentioning, as if a large number of women were excluded on this basis, these results may not accurately reflect the current state of infant wasting in this area. It is also possible that this number if very small, or that all women attended at least 1 ANC visit, but it is still something to consider in my opinion.

3. The first thing discussed in the discussion is the prevalence of wasting in this study compared to other similar areas. Although this is important to note as it highlights the global scale of this issue, I do not think it is the first thing to be discussed here. Instead, I think discussing the findings of this study and highlighting the key factors found to be associated with wasting would be best, as these are the findings that directly address the research question. This could then be tied to a wider discussion of wasting in other geographical areas, providing potential explanations for differences in wasting prevalence (as you have done). In general, I think starting the discussion with study-specific findings and then broadening to the wider context is best.

Overall, this is a considered and well-written report on infant wasting and breastfeeding practices in Ethiopia. With some minor revision, I believe this paper will be a great contribution to the literature.

6. PLOS authors have the option to publish the peer review history of their article (what does this mean? ). If published, this will include your full peer review and any attached files.

**Do you want your identity to be public for this peer review?** For information about this choice, including consent withdrawal, please see our Privacy Policy .

Reviewer #1: No

Reviewer #2: **Yes: ** Maddison Beck

---

## [Author Response · Author response to Decision Letter 1]

15 Nov 2024

Summary of Response to Reviewers Comment

Regarding the study period, data collection for this study took place in Ethiopia during the Ethiopian calendar year 2013. The Ethiopian calendar system is approximately seven to eight years behind the Gregorian calendar. This difference arises due to an alternate calculation of the birth year of Jesus Christ used in the Ethiopian calendar. The Ethiopian year begins on Meskerem 1, which typically falls on September 11 in the Gregorian calendar (or September 12 in a leap year). The initial portion of Ethiopian year 2013 (from Meskerem, which starts in September, through December) corresponds to September–December 2020 in the Gregorian calendar. The latter part of Ethiopian year 2013 (from January through August) aligns with January–August 2021 in the Gregorian calendar. Consequently, while the dataset indicates an Ethiopian calendar year of 2013, it actually encompasses a timeframe that includes late 2020 and most of 2021 in the Gregorian system.

The use of a p-value threshold of less than 0.25 for variable selection in multivariable analysis was chosen to ensure potentially relevant predictors are not excluded prematurely. While the standard p-value of 0.05 is often used to determine statistical significance, a more inclusive threshold, such as 0.25, is applied during initial screening in bivariable analysis to capture variables that may not show strong individual associations with the outcome but could be influential when considered together with other predictors (Bursac et al., 2008). This will minimize the risk of excluding important variables that could improve the model's predictive ability and robustness in multivariable contexts. This statement is also included in the revised manuscrpt on the data analysis part. “All the variables with a p-value of less than 0.25 were used during initial screening in the bivariable analysis to capture variables that may not show strong individual associations with the outcome but could be influential when considered together with other predictors. (Zoran B., et al 2008).

Additionally, we included the prevalence of exclusive breastfeeding in the study area, in the discussion part. “According to the 2019 Ethiopia Demographic and Health Survey (EDHS), the prevalence of exclusive breastfeeding for infants under 6 months in the SNNPR, 58%. In contrast, the rate in the Somali Region is lower, at around 25%.” (EDHS, 2019)

Reviewer Comments

1. Take seriously the following major concerns regarding the data periods claimed. The dates of birth in the dataset are all in 2013 but the study claims to have been carried out in 2021. Justify this discrepancy.

Response: “This concern has been addressed in the summary section provided above.”

2. Along with global statistics, mention the statistics of exclusively breastfeeding in the study area.

Corrected: Yes, “This concern has been addressed in the summary section provided above.”

3. From the Study design and setting, delete the sentence “The study aimed to strengthen nutrition interventions integrated into existing antenatal care services and to improve maternal nutrition practices.

Corrected: Yes, this sentence is deleted from the Study design and setting. “The study aimed to strengthen nutrition interventions integrated into existing antenatal care services and to improve maternal nutrition practices.”

4. The authors write “All the variables with a p-value of less than 0.25 in bivariable analysis were used for multivariable analysis.” What is the reason for using a p-value of less than 0.25? Justify your choice and add a reference(s). Is it “multivariable” or “multivariable” check it carefully. There is a huge difference between them.

Response: Yes, “This concern has been addressed in the summary section provided above.” The revised statement is also included in the revised manuscrpt on the data analysis part.

Thank you for your concern, the term "multivariable" refers to models involving multiple predictors for a single outcome, whereas "multivariate" is used for models with multiple outcomes. In this analysis, "multivariable" is appropriate since we are focusing on multiple independent variables predicting one outcome.

5. It is recommended to show the p-values in bivariate analysis.

Corrected: Yes, p-values in bivariate analysis is included.

6. Limitations of the study should be a separate section.

Corrected: Yes, the limitation is written in the separate section.

Reviewer #1: Thank you for the opportunity to review your work. The underlying approach is sound and many of the comments below are about making the work easier to read, or more precise. However, there is a major concern re. data periods claimed. The dates of birth in the dataset are all in 2013 but the study claims to have been carried out in 2021. It is for this reason I have recommended major revision.

MAJOR POINTS

INTRO

Authors take a long time to state this aim of this study. Suggest adding one or two lines to the end of the first paragraph that states the aim of this work was to look at the association between breastfeeding practices and wasting in infants aged under six months in the South of Ethiopia.

Corrected: Yes, corrected in the manuscript as follows, “the aim of this study was to examine the association between breastfeeding practices and wasting in infants under six months of age in the SNNPR and Somali regions of Ethiopia.”

METHODS

Data claims to be from 2021 but the included dataset has all of the children's date of birth in 2013.

Response: “This concern has been addressed in the summary section provided above.”

RESULTS

Authors discuss LAZ and WAZ in Intro/Results but never refer to it in the manuscript again. Remove any mention of these variables if they are not discussed in detail outside of Table 3.

Corrected: Yes, we removed LAZ and WAZ from the Introduction and Results sections.

DISCUSSION

There is no discussion of null or negative findings. For example, a wealth index is included but is not addressed in the discussion. Similarly, the number of pregnancies and births should be discussed at least briefly as to why they do or not affect the results.

Corrected: Yes, On the discussion part the null or negative findings were discussed. This statement is included in the discussion part as follows. “While the wealth index showed a significant association with infant wasting in the bivariate analysis, this association did not hold in the multivariable analysis when other factors such as breastfeeding practices and maternal education are considered. One possible explanation is that wealth may indirectly influence wasting by affecting these more proximal factors, such as the mother's education level and access to health information, which in turn influence infant feeding practices (Hammond YA, et al., 2024). When these factors are controlled for in the multivariable analysis, the direct association between wealth and wasting diminishes. Additionally, wealth may capture general household resources but may not fully reflect specific nutritional or healthcare practices crucial for preventing infant wasting. Thus, while wealthier households might have better resources, the actual practices around infant feeding and care are key determinants of infant nutrition are more directly relevant to wasting outcomes (Lijalem MT et al 2021). This analysis also explored whether the number of pregnancies and births (parity) was associated with infant wasting. Although higher parity is often thought to affect child health outcomes due to resource allocation or maternal experience, our findings did not show a significant association between the number of pregnancies, parity and wasting in the multivariable analysis.

One possible reason for this non-significance is that factors more directly tied to the immediate care and feeding of the infant, such as exclusive breastfeeding and early initiation of breastfeeding, may be stronger determinants of wasting. The mother’s experience with multiple births may increase her general caregiving knowledge but does not necessarily ensure the specific practices that prevent wasting. Furthermore, higher parity can sometimes strain household resources and maternal time, but these factors may not directly impact infant nutritional status if the household or mother is already practicing adequate breastfeeding practices. (Renata O.N, et al., 2021).

MINOR POINTS

Abstract

Line 26 - Suggest removing "inadequate".

Corrected: Yes, "inadequate" has been removed.

Line 27 - South East, not Southeast.

Corrected: Yes, to ensure clarity, I revised the text to explicitly mention both the SNNPR and Somali Regions in the manuscript.

Line 27 - SNNPR is in the South West of the country. Describing the work as being set only in South East is incorrect.

Corrected: Yes, corrected as discussed above.

Introduction

Line 75 - Capitalisation of World Health Organization

Corrected: Yes, corrected to “World Health Organization”.

Lines 80-84 - This paragraph does not add to the value of the introduction. Consider removing it because the introduction is very long already.

Corrected: Yes, removed the paragraph written in Lines 80-84.

Line 87 - "The prevalence of wasting..." should not be a new sentence but should carry on from the first sentence in the paragraph.

Corrected: Yes, corrected by combining the sentences as follows. “Despite the implementation of various strategies and programs, including the National Nutrition Program (NNP), the prevalence of wasting among children under five remains high in Ethiopia.”

Line 92 - This paragraph starts by mentioned underlying factors but goes on to discuss problems with measurement methodologies. Consider revising first sentence. Line 102 onwards seems to discuss underlying factors but appears to be a new paragraph.

Corrected: Yes, the paragraph stated in Line 92 is removed and discussed the underlying factors in Line 110.

Line 99 - WLZ initialism was defined earlier but "Weight-for-Length Z-score" is written out in full here.

Corrected: Yes, "Weight-for-Length Z-score" changed to “WLZ”.

Methods

Line 124 and elsewhere - It is unclear what the "end-line" survey is.

Response: The study had a base-line and end-line household survey on a large feasibility and impact study of maternal nutrition interventions. For the current study I used the end-line household survey data.

Line 130 - Data period claims to be 2021 but dates of births in dataset are 2013.

Response: “This concern has been addressed in the summary section provided above.”

Line 149 - Were all children singleton births? Multiple births can have lower birth weight which may be relevant to growth comparisons in the early months of life.

Response. "Thank you for raising this important point. While we acknowledge that multiple births can influence birth weight and subsequent growth comparisons, but this was not flagged as a significant issue during data collection, and the data collectors did not report encountering such cases. Additionally, our dataset does not include specific information on whether children were singleton or multiple births, which limits our ability to address this factor directly in the analysis. We appreciate your understanding and will consider this variable in future studies."

Line 154 - It is unclear what is meant by "enumerator training and field practice".

Response: In this study, we used secondary data, so the term 'enumerator training and field practice' refers to the preparation and practices undertaken by the original data collection team. In the primary data collection phase, enumerators were trained to ensure standardized and accurate data collection, and they conducted field practice to familiarize themselves with the data collection tools and procedures before gathering data in real settings. This training and practice aimed to improve data reliability and consistency.

Line 171 - Is zcore06 function from a third-party package? If so, please include a citation.

Corrected: Yes, the zscore06 function is part of a user-written Stata package, specifically designed for calculating child growth z-scores using WHO standards. We used Stata version 17, and we have included a citation in the manuscript. (Leroy JL. (2011)

Line 175 - LAZ and WAZ do not appear to be mentioned anywhere else in the article to please remove this line.

Corrected: Yes, LAZ and WAZ removed from the manuscript.

Line 178 - Was mother height (included in the supplied dataset) used in any of the models? If so, was it the continuous variable or the categorical in the dataset?

Response: Yes, it was included in the supplied dataset, but it was not used in any of the models, because the p-value is above 0.25 in the bivariate analysis.

The maternal height is categorized in to three by using a command, xtile mother-heigh = height1, nq (3)

Mother-heigh was labeled as 1 "short" 2 "medium" 3 "tall". It was included in the explanatory variable of the manuscript.

Line 183 - Was the "child perceived birth size" variable used anywhere? If so, this should be recognized as a limitation since it is subject to recall bias and a subjective variable.

Response: The child perceived birth size variable is described in numbers in Table 3. But not included in any of the model because in the bivariate analysis the p-value is above 0.25.

Line 191 - What is an "index child"? No definition given.

Corrected: Yes, this definition is included in the explanatory variables. “The index child is the specific child selected for the study during data collection, chosen from among the children available in the household.”

Line 192 - Use semi-colons to delineate between items in the list.

Corrected: Yes, semi-colons were used to delineate between items.

Line 210 - The citation for the wealth index is a technical report but I couldn't find where it lists the consumer items. A more in-depth explanation of how the wealth index is calculated in this study is required. This may be a supplemental file that lists the "20 items" that the authors describe, and how they used these to divide in to quintiles.

Corrected: Yes, twenty-one variables were used to develop the wealth index, and included as a supplemental file by listing all the twenty-one variables.

To clarify the calculation steps for the wealth index, here is the procedure by using Stata commands:

1. Conducting PCA and Checking Sampling Adequacy

First, we used Principal Component Analysis (PCA) to create a composite measure of wealth. To ensure the dataset was adequate for PCA, we checked the Kaiser-Meyer-Olkin (KMO) measure of sampling adequacy: estat kmo

2. Extracting the First Component

We generated the first principal component from the PCA results as an asset score, which represents the wealth index score: predict comp1 rename comp1 asset_scor

3. Creating Wealth Quintiles

Next, we divided the asset score into quintiles (five equal groups), with each group representing different wealth levels. This was done using the xtile command, which created the asset_inde variable with values ranging from 1 to 5: xtile asset_inde = asset_scor, nq(5)

4. Labeling the Wealth Categories

Finally, we labeled each quintile to define wealth categories from "poor" to "richer." We used the recode and label define commands to assign descriptive labels to each group: recode asset_inde (1=1 "poor") (2=2 "poorer") (3=3 "middle") (4=4 "rich") (5=5 "richer"), gen(Wealth_index) label define asset_inde 1 "poor" 2 "poorer" 3 "middle" 4 "rich" 5 "richer", replace

Line 237 - Suggest adding "only", e.g. "Model I was adjusted for 238 socio-demographic variables ONLY, Model II was adjusted for maternal-related variables ONLY..."

Corrected: Yes, corrected as follows, “Model I was adjusted for socio-demographic variables only, Model II was adjusted for maternal-related variables only, Model III was adjusted for infant-related variables only.”

Methods general note - Explain that there may be some uncertainty around the child's day of birth so some ages were given to the nearest month, only. Explain how missing or ambiguous data were handled.

Corrected: Yes, this statement is included in the data collection procedure part as follows. “For the original st

---

## [Decision Letter · Decision Letter 1]

9 Dec 2024

PONE-D-24-29945R1Association Between Wasting and Inadequate Breastfeeding Practices Among Infants Under Six Months in Southeast Ethiopia: A Multilevel Cross-Sectional StudyPLOS ONE

Dear Dr. Getachew,

Thank you for submitting your manuscript to PLOS ONE. After careful consideration, we feel that it has merit but does not fully meet PLOS ONE’s publication criteria as it currently stands. Therefore, we invite you to submit a revised version of the manuscript that addresses the points raised during the review process.

We look forward to receiving your revised manuscript.

Kind regards,

Md. Moyazzem Hossain, PhD

Academic Editor

PLOS ONE

Journal Requirements:

Reviewers' comments:

Reviewer's Responses to Questions

**Comments to the Author**

1. If the authors have adequately addressed your comments raised in a previous round of review and you feel that this manuscript is now acceptable for publication, you may indicate that here to bypass the “Comments to the Author” section, enter your conflict of interest statement in the “Confidential to Editor” section, and submit your "Accept" recommendation.

Reviewer #1: (No Response)

2. Is the manuscript technically sound, and do the data support the conclusions?

Reviewer #1: Yes

3. Has the statistical analysis been performed appropriately and rigorously? 

Reviewer #1: Yes

4. Have the authors made all data underlying the findings in their manuscript fully available?

Reviewer #1: Yes

5. Is the manuscript presented in an intelligible fashion and written in standard English?

Reviewer #1: No

6. Review Comments to the Author

Reviewer #1: Thank you for addressing many of the comments made on the previous version of the manuscript. Although there are a lot of comments below, they are nearly all about making the manuscript easier to read, more consistent, and grammatically correct. I really look forward to seeing this published and my comments are sincerely intended to make this article a more cleanly-polished final version.

(These notes refer to the updated manuscript beginning on p.98 of the reviewer PDF download. This is the version of the manuscript which displays edits in red and blue throughout.)

# GENERAL

# ABSTRACT

Line 23 - Subsection is Background but main body of paper uses Introduction

Line 27 - "...age in ^THE SNNPR..."

Line 30 - Authors have used "Southern". Please use only "SNNPR" notation throughout for consistency.

Line 32 - I would drop the line about STATA from the abstract. Irrelevant detail for an abstract.

Line 37 - Trailing zero missing from "1.5".

Line 38 - Inconsistent level of precision 1.52 then 1.0, 2.3. Please change to use a consistent number of decimal places.

Line 39 - Trailing zero mnissing from "1.5"

Line 40 - Missing trailing zeroes on 0.4 and 0.3

Line 41 - Suggest adding some words to clarify if "...independently associated with wasting" mean those variables were significant during multivariable regression or just in bivariable. I presume the authors mean the multivariate regression but please make this clearer.

Line 42 - Subsection is Conclusion but main body of paper uses Discussion

Line 45 - "may help" rather than "can help". This study was not an intervention so cannot make that claim, only suggest.

# INTRODUCTION

Line 65 - First paragraph should include a brief description of the study aim. Authors have reworded the final paragraph of the Introduction but have not added this to the first paragraph, as requested in the previous review. Having to read multiple pages of introduction without knowing the purpose of the study is frustrating.

Line 75 - "deviation" should be plurarlised to "deviations"

Line 94 - Were the 9.4% of children with wasting at the national level or within the regions of interest?

Line 102 - This should be changed back to "Additionally, the mid-upper arm circumference IS not typically for this age group."

Line 107 - I don't think this should be a new paragraph.

Line 122 - Consider mentioning that end-line data were used, since it was taken from an intervention study.

Line 145 - Replace with Southern with SNNPR for consistency.

Line 148 - Does Southern mean SNNPR?

Line 156 - Does Southern mean SNNPR?

# METHODS

Line 190 - "The ^PRIMARY study outcome..." since mutiple outcomes are discussed.

Line 207 - I don't think a new sentence is needed at "The age of the infant...". If so, then perceived birth size should also be a new setence.

Line 210 - "Infant feeding indicators WERE..." Consistence use of tense.

Line 214 - Suggest "The following indicators were calculated..."

Line 214 - Inconsistent use of capitalisation in this paragraph. Previous paragraphs do not capitalise items in the list. Suggest that "Infant ever...", "Timing of..." etc do not need capital letters.

Line 225 - As above, inconsistent use of capital letters and list make this difficult to follow for the reader.

Line 231 - Missing period at the end of sentence.

Line 232 - Citation for wealth index should be given in this sentence, rather than the end of paragraph.

Line 246 - "Descriptive statistics WERE..."

Line 249 - Suggest removing extended explanation of 95% CI's "...providing a range within which 250 the true population parameters are expected to lie with 95% confidence, adding 251 insight into the precision and reliability of our estimates."

Line 259 - "...significance WAS declared..."

# RESULTS

Line 295 - Parentheses around 67% and 32.8% would be appropriate.

Line 296 - Inconsistent precision 67% and 32.8% use different number of decimal places.

Line 296 - Parentheses around 63.% and 36.9% would be appropriate.

Line 297 - Inconsistent precision 63% and 36.9% use different number of decimal places.

Line 304 - Add trailing zeroes where needed. For example, "Middle" should be 20.0, not 20. Please check all rows for this issue.

Line 308 - Inconsistent precision 42% has fewer decimal places compared to rest of paragraph.

Line 319 - Add trailing zeroes where needed. For example, "3-6 month" should be 61.0, not 61. Please check all rows for this issue.

Line 330 - "Exclusive breastfeeding status..." should be in the Methods section?

Line 335 - Add trailing zeroes where needed. For example, "Stunted Yes" should be 10.0, not 10. Please check all rows for this issue.

Line 340 - Inconsistent precision 58% has fewer decimal places than 41.8%.

Line 342 - Change tense throughout "...wasting WAS higher at FIVE months..." rather than IS.

Line 342 - Numbers of nine or less should be expressed as words. "...wasting WAS higher at five months...". Change throughout paragraph.

Line 348 - It is unclear what other category there may be excepted wasted or not wasted. If 26.2% of children are wasted at five months and 24.7% of children are not wasted at five months, what has happened to the remaining ~50%?

Line 356 - Parentheses around 66.7% would be appropriate.

Line 360 - Remove "suggesting that higher maternal education is associated with a lower prevalence of child wasting." This is a point for Discussion, not Results.

Line 374 - Missing trailing zeroes in some rows. For eample, Prevalence of "Secondary and above education" is given as "5(1,13.9)" but should be given as "5.0 (1.0, 13.9)".

Line 374 - Change column header to "Prevalence (95% CI)" with parentheses

Line 374 - Spaces should be placed between values and parentheses to make this table easier to read. For example, "20-24" in the Prevalence column - 14.2(9.3,20.3) should be 14.2 (9.3, 20.3). All of these values in the table should be formatted as such for clarity.

Line 395 - As above. Please add spaces to make values more easily readable throughout this table.

Line 395 - Hyphens used for 95% CI in this table but commas have been used in Table 4. Please go through the manuscript and choose one notation and use consistently.

Line 395 - Suggest "Child age (months)" to make it clear it is measured in months.

# DISCUSSION

Line 404 - "Socio-demographic" does not need a capital letter and no capital is used on Line 584 for the same phrase.

Line 404 - Add "... in the Somali and SNNPR regions of Ethopia"

Line 411 - I appreciate the addition of the reference to the regional statistics as per my suggestion of my previous review. But as the authors point out, this talks about a different age group. I was wrong to suggest this comparison. I leave it to the authors to decide if they wish to remove this comparison.

Line 418 - Inconsistent precision 14.2% and 16% use different numbers of decimal places.

Line 423 - As per Line 418 comment.

Line 445 - Clarify if 1.48 or 1.50.

Line 448 - What rate did Negatu et al observe, specifically? Same for studies in Nigeria and Bangladesh. Please report the ratios from these studies here.

Line 473 - As above, please report the specific ratios found in citations number 44 and 45 directly in the Discussion.

Line 476 - "...HAVING a protective..."

Line 508 - What ratio did citation 19 find for sex differences? Please report directly here so the reader can see if it compares to the authors' value of 1.50.

Line 510 - As above for citation 51.

Line 581 - "StrengthS" rather than "Strength"

Line 592 - "LimitationS" rather than "Limitation"

Line 592 - Add a brief note to explain that information about singleton/multiple births was not collected.

Line 602 - Please use either SNNPR or Southern throughout the paper (Ctrl + F!)

7. PLOS authors have the option to publish the peer review history of their article (what does this mean? ). If published, this will include your full peer review and any attached files.

**Do you want your identity to be public for this peer review?** For information about this choice, including consent withdrawal, please see our Privacy Policy .

Reviewer #1: No

---

## [Author Response · Author response to Decision Letter 2]

20 Dec 2024

Summary of Response to Reviewers Comment

Based on the reviewers' comments, the following revisions were made:

1. The first paragraph was revised as follows: “This study aims to investigate the relationship between breastfeeding practices and wasting among infants under six months of age in Ethiopia, addressing a critical gap in understanding wasting in this vulnerable age group.”

2. Throughout the manuscript, the term "Southern" was replaced with "SNNPR" for consistency.

3. In Tables 4 and 5, spaces were added between values and parentheses, and commas were used to improve clarity.

4. The level of precision for all numerical data in the manuscript, both in text and tables, was standardized and corrected.

5. For the comment raised in Line 348, the explanation is as follows, the percentages for wasted and non-wasted children were calculated independently within each age group. For wasted children, the percentages represent their distribution within the total number of wasted children across all age groups, summing to 100%. Similarly, for non-wasted children, the percentages represent their distribution within the total number of non-wasted children, also summing to 100%. These calculations were done separately for wasted and non-wasted categories, and the percentages are not combined into a single total. To address any potential confusion, the sentence in the manuscript was reworded for clarity.

Reviewer Comments

Reviewer #1

General Comments

ABSTRACT

Line 23 - Subsection is Background but main body of paper uses Introduction

Corrected: Yes, corrected in the manuscript as follows, “Introduction changed to Background”

Line 27 - "...age in ^THE SNNPR..."

Corrected: Yes, corrected as follows, “in the SNNPR and Somali regions of Ethiopia”.

Line 30 - Authors have used "Southern". Please use only "SNNPR" notation throughout for consistency.

Corrected: Yes, corrected, "Southern", changed in to "SNNPR".

Line 32 - I would drop the line about STATA from the abstract. Irrelevant detail for an abstract.

Corrected: Yes, removed the word written in line 32.

Line 37 - Trailing zero missing from "1.5".

Corrected: Yes, corrected by adding zero, 1.5 to 1.50

Line 38 - Inconsistent level of precision 1.52 then 1.0, 2.3. Please change to use a consistent number of decimal places.

Corrected: Yes, corrected by adding the numbers as follows, 1.52; CI:1.00, 2.30.

Line 39 - Trailing zero mnissing from "1.5"

Corrected: Yes, corrected by adding zero, 1.5 to 1.50

Line 40 - Missing trailing zeroes on 0.4 and 0.3

Corrected: Yes, corrected by adding zero, “0.40 and 0.30”

Line 41 - Suggest adding some words to clarify if "...independently associated with wasting" mean those variables were significant during multivariable regression or just in bivariable. I presume the authors mean the multivariate regression but please make this clearer.

Corrected: Yes, corrected as follows, “were independently associated with wasting in the multivariable analysis”.

Line 42 - Subsection is Conclusion but main body of paper uses Discussion

Thank you for your valuable feedback. We acknowledge the inconsistency in terminology between the "Conclusion" subsection and the use of "Discussion" in the main body of the paper. In preparing the abstract, we followed the PLOS ONE format, which is unstructured and typically includes only the background, methods, results, and conclusions. This format is widely accepted and ensures clarity, conciseness, and alignment with PLOS ONE's submission guidelines.

Line 45 - "may help" rather than "can help". This study was not an intervention so cannot make that claim, only suggest.

Corrected: Yes, corrected by changing in to "may help".

# INTRODUCTION

Line 65 - First paragraph should include a brief description of the study aim. Authors have reworded the final paragraph of the Introduction but have not added this to the first paragraph, as requested in the previous review. Having to read multiple pages of introduction without knowing the purpose of the study is frustrating.

Corrected: Yes, the first paragraph was revised to include the aim of the study at the end, as follows: “This study aims to investigate the relationship between breastfeeding practice and wasting among infant under six months of age in Ethiopia, addressing a critical gap in understanding wasting in this vulnerable age group.”

Line 75 - "deviation" should be plurarlised to "deviations"

Corrected: Yes, corrected by adding “s”, "deviations"

Line 94 - Were the 9.4% of children with wasting at the national level or within the regions of interest?

Corrected: Yes, corrected as follows, “9.4% of infants under six months of age were identified as wasted at the national level.”

Line 102 - This should be changed back to "Additionally, the mid-upper arm circumference IS not typically for this age group."

Corrected: Yes, corrected as follows, “Additionally, the mid-upper arm circumference is not typically for this age group.”

Line 107 - I don't think this should be a new paragraph.

Corrected: Yes, corrected

Line 122 - Consider mentioning that end-line data were used, since it was taken from an intervention study.

Corrected: Yes, corrected as follows “This analysis was conducted using the end-line household survey data from a maternal nutrition intervention study.”

Line 145 - Replace with Southern with SNNPR for consistency.

Corrected: Yes, corrected, "Southern", changed in to "SNNPR".

Line 148 - Does Southern mean SNNPR?

Corrected: Yes, corrected, "Southern", changed in to "SNNPR".

Line 156 - Does Southern mean SNNPR?

Corrected: Yes, corrected, "Southern", changed in to "SNNPR".

# METHODS

Line 190 - "The ^PRIMARY study outcome..." since mutiple outcomes are discussed.

Corrected: Yes, corrected by adding, “The primary study outcome variable was wasting among infants under six months of age”

Line 207 - I don't think a new sentence is needed at "The age of the infant...". If so, then perceived birth size should also be a new sentence.

Corrected: Yes, corrected.

Line 210 - "Infant feeding indicators WERE..." Consistence use of tense.

Corrected: Yes, corrected by changing "are" to "were."

Line 214 - Suggest "The following indicators were calculated..."

Corrected: Yes, corrected by adding “the following indicators were”

Line 214 - Inconsistent use of capitalisation in this paragraph. Previous paragraphs do not capitalise items in the list. Suggest that "Infant ever...", "Timing of..." etc do not need capital letters.

Corrected: Yes, corrected as follows, “infant ever breastfeeds; timing of breastfeeding initiation; colostrum feeding; pre-lacteal feeding; bottle feeding; started cow/goat/camel milk; started solid, semi-solid, and soft food were coded as "no" and "yes."

Line 225 - As above, inconsistent use of capital letters and list make this difficult to follow for the reader.

Corrected: Yes, revised by changing the capital letters to lowercase.

Line 231 - Missing period at the end of sentence.

Corrected: Yes, revised by adding a period at the end of the sentence.

Line 232 - Citation for wealth index should be given in this sentence, rather than the end of paragraph.

Corrected: Yes, the citation for wealth index was given.

Line 246 - "Descriptive statistics WERE..."

Corrected: Yes, revised by changing "was" to "were."

Line 249 - Suggest removing extended explanation of 95% CI's "...providing a range within which 250 the true population parameters are expected to lie with 95% confidence, adding 251 insight into the precision and reliability of our estimates."

Corrected: Yes, corrected as follows, The prevalence of wasting was determined using proportions with 95% confidence intervals (CIs), to indicate the precision of the estimates.

Line 259 - "...significance WAS declared..."

Corrected: Yes, revised by changing "were" to "was."

# RESULTS

Line 295 - Parentheses around 67% and 32.8% would be appropriate.

Corrected: Yes, corrected by adding parentheses, (67.2%) and (32.8%)

Line 296 - Inconsistent precision 67% and 32.8% use different number of decimal places.

Corrected: Yes, corrected as follows, (67.2%) and (32.8%)

Line 296 - Parentheses around 63.% and 36.9% would be appropriate.

Corrected: Yes, corrected by adding parentheses, (63.1%) and (36.9%).

Line 297 - Inconsistent precision 63% and 36.9% use different number of decimal places.

Corrected: Yes, corrected as follows, “(63.1%) and (36.9%).”

Line 304 - Add trailing zeroes where needed. For example, "Middle" should be 20.0, not 20. Please check all rows for this issue.

Corrected: Yes, the numbers in Table 1 were updated as follows: "Middle, Rich, and Richer" were changed to 20.0.

Line 308 - Inconsistent precision 42% has fewer decimal places compared to rest of paragraph.

Corrected: Yes, revised by changing "42%" to "42.1%"

Line 319 - Add trailing zeroes where needed. For example, "3-6 month" should be 61.0, not 61. Please check all rows for this issue.

Corrected: Yes, revised by changing the numbers in Table 2, “61 to 61.0”.

Line 330 - "Exclusive breastfeeding status..." should be in the Methods section?

Corrected: Yes, revised by removing the sentence, as the "Exclusive breastfeeding status" is already discussed in the methods section.

Line 335 - Add trailing zeroes where needed. For example, "Stunted Yes" should be 10.0, not 10. Please check all rows for this issue.

Corrected: Yes, revised by changing "10" to "10.0"

Line 340 - Inconsistent precision 58% has fewer decimal places than 41.8%.

Corrected: Yes, revised by changing "58%" to "58.2%"

Line 342 - Change tense throughout "...wasting WAS higher at FIVE months..." rather than IS.

Corrected: Yes, corrected as follows, “The proportion of wasted children varies with age, wasting was higher at five months, where 26.2% of wasted cases are observed. Wasting was lowest at one months, 8.5% and shows an increase at two months, 20.6% remaining relatively high from two to five months.

Line 342 - Numbers of nine or less should be expressed as words. "...wasting WAS higher at five months...". Change throughout paragraph.

Corrected: Yes, corrected as follows, “The proportion of wasted children varies with age, wasting was higher at five months, where 26.2% of wasted cases are observed. Wasting was lowest at one months, 8.5% and shows an increase at two months, 20.6% remaining relatively high from two to five months.”

Line 348 - It is unclear what other category there may be excepted wasted or not wasted. If 26.2% of children are wasted at five months and 24.7% of children are not wasted at five months, what has happened to the remaining ~50%?

Corrected: Yes, discussed in the summary of the response to the reviewer, the percentages for wasted and non-wasted children were calculated independently within each age group. For wasted children, the percentages represent their distribution within the total number of wasted children across all age groups, summing to 100%. Similarly, for non-wasted children, the percentages represent their distribution within the total number of non-wasted children, also summing to 100%. These calculations were done separately for wasted and non-wasted categories, and the percentages are not combined into a single total. To address any potential confusion, the sentence in the manuscript was reworded for clarity.

Line 356 - Parentheses around 66.7% would be appropriate.

Corrected: Yes, corrected by adding parentheses.

Line 360 - Remove "suggesting that higher maternal education is associated with a lower prevalence of child wasting." This is a point for Discussion, not Results.

Corrected: Yes, revised by removing the sentence

Line 374 - Missing trailing zeroes in some rows. For eample, Prevalence of "Secondary and above education" is given as "5(1,13.9)" but should be given as "5.0 (1.0, 13.9)".

Corrected: Yes, revised by adding trailing zeroes in Table 4, for example, "5(1,13.9)" was changed to “5.0 (1.0, 13.9)”.

Line 374 - Change column header to "Prevalence (95% CI)" with parentheses

Corrected: Yes, corrected as follows, "Prevalence (95% CI)".

Line 374 - Spaces should be placed between values and parentheses to make this table easier to read. For example, "20-24" in the Prevalence column - 14.2(9.3,20.3) should be 14.2 (9.3, 20.3). All of these values in the table should be formatted as such for clarity.

Corrected: Yes, revised by adding spaces between values and parentheses in Table 4.

Line 395 - As above. Please add spaces to make values more easily readable throughout this table.

Corrected: Yes, corrected by adding space between values and parentheses

Line 395 - Hyphens used for 95% CI in this table but commas have been used in Table 4. Please go through the manuscript and choose one notation and use consistently.

Corrected: Yes, revised by using commas consistently in both Table 4 and Table 5.

Line 395 - Suggest "Child age (months)" to make it clear it is measured in months.

Corrected: Yes, corrected by adding, “Child age (completed months)”

# DISCUSSION

Line 404 - "Socio-demographic" does not need a capital letter and no capital is used on Line 584 for the same phrase.

Corrected: Yes, corrected.

Line 404 - Add "... in the Somali and SNNPR regions of Ethopia"

Corrected: Yes, corrected by adding, in the Somali and SNNPR regions of Ethiopia"

Line 411 - I appreciate the addition of the reference to the regional statistics as per my suggestion of my previous review. But as the authors point out, this talks about a different age group. I was wrong to suggest this comparison. I leave it to the authors to decide if they wish to remove this comparison.

Corrected: Yes, revised by removing the sentence that refers to a different age group.

Line 418 - Inconsistent precision 14.2% and 16% use different numbers of decimal places.

Corrected: Yes, revised by adding a trailing zero, “16.0%”.

Line 423 - As per Line 418 comment.

Corrected: Yes, corrected as follows, “72.2%”

Line 445 - Clarify if 1.48 or 1.50.

Corrected: Yes, corrected. The prevalence of the odds ratio was 1.50, the value 1.48 was a typographic error.

Line 448 - What rate did Negatu et al observe, specifically? Same for studies in Nigeria and Bangladesh. Please report the ratios from these studies here.

Correction: Yes, corrected by including the prevalence of the odds ratio for the studies by Negatu et al. and Bangladesh. However, for the study conducted in Nigeria, the prevalence was not clearly specified.

Line 473 - As above, please report the specific ratios found in citations number 44 and 45 directly in the Discussion.

Corrected: Yes, corrected as follows, “This study found that infants who did not initiate breastfeeding within one hour of birth had 52% higher odds of wasting compared to those who started breastfeeding within the first hour. This finding aligns with research conducted across 20 developing countries using demographic health survey data, which reported a 31% increase in the odds of wasting among infants who delayed breastfeeding initiation [40]. Similarly, a study from South Asia demonstrated that early breastfeeding initiation reduced the odds of wasting by 8% [41]. These findings highlight the importance of timely breastfeeding initiation in reducing the risk of wasting across various populations.”

Line 476 - "...HAVING a protective..."

Corrected: Yes, corrected as follows, “This could be due to early initiation of breastfeeding having a protective effect against infections and reducing newborn mortality.”

Line 508 - What ratio did citation 19 find for sex differences? Please report directly here so the reader can see if it compares to the authors' value of 1.50.

Corrected: Yes, corrected as follows, “This study showed that males were 50% more likely to be wasted than females. This finding aligns with data from the 2019 Ethiopian Mini Demographic Health Survey, which reported that being female was associated with 30% lower odds of wasting [15]. Similarly, a study conducted in three disadvantaged East African Districts, Rwanda, Uganda, and Tanzania among children under five years of age found that being female reduced the odds of wasting by 14% [47].”

Line 510 - As above for citation 51.

Corrected: Yes, corrected.

Line 581 - "StrengthS" rather than "Strength"

Cor

---

## [Decision Letter · Decision Letter 2]

30 Dec 2024

PONE-D-24-29945R2Association Between Wasting and Inadequate Breastfeeding Practices Among Infants Under Six Months in SNNPR and Somali regions of Ethiopia: A Multilevel Cross-Sectional StudyPLOS ONE

Dear Dr. Getachew,

Thank you for submitting your manuscript to PLOS ONE. After careful consideration, we feel that it has merit but does not fully meet PLOS ONE’s publication criteria as it currently stands. Therefore, we invite you to submit a revised version of the manuscript that addresses the points raised during the review process.

We look forward to receiving your revised manuscript.

Kind regards,

Md. Moyazzem Hossain, PhD

Academic Editor

PLOS ONE

Journal Requirements:

Reviewers' comments:

Reviewer's Responses to Questions

**Comments to the Author**

1. If the authors have adequately addressed your comments raised in a previous round of review and you feel that this manuscript is now acceptable for publication, you may indicate that here to bypass the “Comments to the Author” section, enter your conflict of interest statement in the “Confidential to Editor” section, and submit your "Accept" recommendation.

Reviewer #1: (No Response)

2. Is the manuscript technically sound, and do the data support the conclusions?

Reviewer #1: Yes

3. Has the statistical analysis been performed appropriately and rigorously? 

Reviewer #1: Yes

4. Have the authors made all data underlying the findings in their manuscript fully available?

Reviewer #1: Yes

5. Is the manuscript presented in an intelligible fashion and written in standard English?

Reviewer #1: No

6. Review Comments to the Author

Reviewer #1: Dear authors,

Thank you for perservering and making the many corrections I have previously suggested. There are still some outstanding issues to address in the Results section. These corrections are, again, about making the paper easier to read. Since PLOS One does not provide any copyedit function, I have tried to correct all typos and grammar issues.

I would like to say again that I really can't wait to see this work published. I hope you understand that my suggested corrections are about making this maunscript properly showcase the good work you have carried out.

Note these comments are based on R2 which starts on Page 174/246 of the reviewer PDF.

# Abstract

All changes implemented from previous review. The abstract is now more consistent and has better grammar/readability.

# Introduction

Purpose of study being included in the first paragraph is a significant improvement, thank you.

Line 99 - "...used..." has been deleted but I think a word like "used" was meant to replace it? You could probably just delete this sentence since it distracts the reader, if you want.

# Methods

It looks like all grammar and punctuation issues have been fixed. The section is now easier to read. Thanks!

# Results

Line 280 - Minor point: sentence begins with a numeral. This is not typical. Consider minor rewrite. (Contrast to Line 298.)

Line 282 - "while the REMAINDER (32.8%)..." rather than REMAINING

Line 324 - Suggestion for improved sentence and addition of a missing percentage "Wasting was lowest at age one month (8.5%), increased by age two months (20.6%) and remained relative high from ages two to five months (XX.X%)"

Line 327 - As per the authors' response to the previous review, this paragraph is meant to describe the age DISTRIBUTION of the wasted children, and, separately the age DISTRIBUTION of the non-wasted children so the percentages in the waste group sum to 100% and the percentages in the non-wasted group sum to 100%. Please rewrite this paragraph in a more systematic way. I suggest the following (but with the correct numbers!) "The age distribution of children in the wasted group was 10% age zero months, 20% age one month, 50% aged two to five months and 20% aged over five months. The age distribution of children in the non-wasted group was 30% age zero months, 10% age one month, 20% aged two to five months, and 40% aged over five months."

Line 333 - "A HIGH proportion..."

Line 337 - "...WERE from mothers"

Line 346 - Some spaces have been added but not in all columns. The Wasting Yes/No columns should also have the formatting corrected from, e.g., 55(7.7) to 55 (7.7)

Line 346 - If you are reporting confidence intervals you do not need to report p-values at the same time. The overlap of the confidence intervals demonstrates that, e.g., prevalence of wasting did not vary significantly for any age compared to your reference group (15-19 years).

Line 357 - "Wasting among infants who were not exclusively breastfed was 1.50 times more likely compared to infants who were exclusively breastfed (APR = 1.50; 95% CI:1.02, 2.21)." Deleting some words to make the sentence clearer.

Line 360 - Suggest using exact same phrasing as previous sentence, instead of switching to "52%". "Wasting among infants who initiated breastfeeding more than one hour after birth were 1.52 times more likely compared to infants who start breasteeding less than one hour after birth (APR = 1.52; 95% CI: 1.00, 2.30)."

Consistent levels of precision in tables and text have improved the flow of the writing. Thanks!

# Discussion

Line 384 - "...Barisal district (18.8%)..." - Remove the word "at" before the (18.8%).

Line 433 - Reference needed to support this claim about gastrointestinal diseases. May already be supported by Ref 39 in the next sentence?

7. PLOS authors have the option to publish the peer review history of their article (what does this mean? ). If published, this will include your full peer review and any attached files.

**Do you want your identity to be public for this peer review?** For information about this choice, including consent withdrawal, please see our Privacy Policy .

Reviewer #1: No

---

## [Author Response · Author response to Decision Letter 3]

3 Jan 2025

Response to Reviewers Comment

Thank you for your valuable comments and feedback. Based on your suggestions, the following revisions have been made:

# Abstract

All changes implemented from previous review. The abstract is now more consistent and has better grammar/readability.

# Introduction

Purpose of study being included in the first paragraph is a significant improvement, thank you.

Line 99 - "...used..." has been deleted but I think a word like "used" was meant to replace it? You could probably just delete this sentence since it distracts the reader, if you want.

Corrected: Yes, corrected by removing the sentence.

# Methods

It looks like all grammar and punctuation issues have been fixed. The section is now easier to read. Thanks!

# Results

Line 280 - Minor point: sentence begins with a numeral. This is not typical. Consider minor rewrite. (Contrast to Line 298.)

Corrected: Yes, corrected as follows “More than half of the mothers (68.0%) were housewives, and 52.3% had no formal education.”

Line 282 - "while the REMAINDER (32.8%)..." rather than REMAINING

Corrected: Yes, corrected by changing remaining to reminder.

Line 324 - Suggestion for improved sentence and addition of a missing percentage "Wasting was lowest at age one month (8.5%), increased by age two months (20.6%) and remained relative high from ages two to five months (XX.X%)"

Corrected: Yes, corrected as follows, “The age distribution of children in the wasted group was 11.4% aged zero months, 8.5% aged one month, 20.6% aged two months, and 59.5% aged three to five months. The age distribution of children in the non-wasted group was 9.3% aged zero months, 15.6% aged one month, 20.0% aged two months, and 55.1% aged three to five months."

Line 327 - As per the authors' response to the previous review, this paragraph is meant to describe the age DISTRIBUTION of the wasted children, and, separately the age DISTRIBUTION of the non-wasted children so the percentages in the waste group sum to 100% and the percentages in the non-wasted group sum to 100%. Please rewrite this paragraph in a more systematic way. I suggest the following (but with the correct numbers!) "The age distribution of children in the wasted group was 10% age zero months, 20% age one month, 50% aged two to five months and 20% aged over five months. The age distribution of children in the non-wasted group was 30% age zero months, 10% age one month, 20% aged two to five months, and 40% aged over five months."

Corrected: Yes, corrected as follows, " The age distribution of children in the wasted group was 11.4% aged zero months, 8.5% aged one month, 20.6% aged two months, and 59.5% aged three to five months. The age distribution of children in the non-wasted group was 9.3% aged zero months, 15.6% aged one month, 20.0% aged two months, and 55.1% aged three to five months."

Line 333 - "A HIGH proportion..."

Corrected: Yes, corrected by changing higher to high.

Line 337 - "...WERE from mothers"

Corrected: Yes, corrected by changing “are to where”.

Line 346 - Some spaces have been added but not in all columns. The Wasting Yes/No columns should also have the formatting corrected from, e.g., 55(7.7) to 55 (7.7)

Corrected: Yes, space was added to the Wasting Yes/No columns.

Line 346 - If you are reporting confidence intervals you do not need to report p-values at the same time. The overlap of the confidence intervals demonstrates that, e.g., prevalence of wasting did not vary significantly for any age compared to your reference group (15-19 years).

Corrected: Yes, the p-values reported were removed.

Line 357 - "Wasting among infants who were not exclusively breastfed was 1.50 times more likely compared to infants who were exclusively breastfed (APR = 1.50; 95% CI:1.02, 2.21)." Deleting some words to make the sentence clearer.

Corrected: Yes, "Infants who were not exclusively breastfed were 1.50 times more likely to experience wasting compared to those who were exclusively breastfed (APR = 1.50; 95% CI: 1.02–2.21)."

Line 360 - Suggest using exact same phrasing as previous sentence, instead of switching to "52%". "Wasting among infants who initiated breastfeeding more than one hour after birth were 1.52 times more likely compared to infants who start breasteeding less than one hour after birth (APR = 1.52; 95% CI: 1.00, 2.30)."

Corrected: Yes, corrected as follows, “Wasting among infants who initiated breastfeeding more than one hour after birth were 1.52 times more likely compared to infants who started breastfeeding less than one hour after birth (APR = 1.52; 95% CI: 1.00, 2.30)."

Consistent levels of precision in tables and text have improved the flow of the writing. Thanks!

# Discussion

Line 384 - "...Barisal district (18.8%)..." - Remove the word "at" before the (18.8%).

Corrected: Yes, by removing “at”

Line 433 - Reference needed to support this claim about gastrointestinal diseases. May already be supported by Ref 39 in the next sentence?

Yes, the sentence was supported by reference 39.

---

## [Decision Letter · Decision Letter 3]

15 Jan 2025

Association Between Wasting and Inadequate Breastfeeding Practices Among Infants Under Six Months in SNNPR and Somali Regions of Ethiopia: A Multilevel Cross-Sectional Study

PONE-D-24-29945R3

Dear Dr. Getachew,

We’re pleased to inform you that your manuscript has been judged scientifically suitable for publication and will be formally accepted for publication once it meets all outstanding technical requirements.

Kind regards,

Md. Moyazzem Hossain, PhD

Academic Editor

PLOS ONE

Additional Editor Comments (optional):

Reviewers' comments:

Reviewer's Responses to Questions

**Comments to the Author**

1. If the authors have adequately addressed your comments raised in a previous round of review and you feel that this manuscript is now acceptable for publication, you may indicate that here to bypass the “Comments to the Author” section, enter your conflict of interest statement in the “Confidential to Editor” section, and submit your "Accept" recommendation.

Reviewer #1: All comments have been addressed

2. Is the manuscript technically sound, and do the data support the conclusions?

Reviewer #1: Yes

3. Has the statistical analysis been performed appropriately and rigorously? 

Reviewer #1: Yes

4. Have the authors made all data underlying the findings in their manuscript fully available?

Reviewer #1: Yes

5. Is the manuscript presented in an intelligible fashion and written in standard English?

Reviewer #1: Yes

6. Review Comments to the Author

Reviewer #1: Thank you for making the many suggested revisions. Your patience has been appreciated. I hope the changes have strengthenede the paper and it does well.

7. PLOS authors have the option to publish the peer review history of their article (what does this mean? ). If published, this will include your full peer review and any attached files.

**Do you want your identity to be public for this peer review?** For information about this choice, including consent withdrawal, please see our Privacy Policy .

Reviewer #1: No

---

## [Editor Report · Acceptance letter]

PONE-D-24-29945R3

PLOS ONE

Dear Dr. Getachew,

I'm pleased to inform you that your manuscript has been deemed suitable for publication in PLOS ONE. Congratulations! Your manuscript is now being handed over to our production team.

Kind regards,

on behalf of

Professor Md. Moyazzem Hossain

Academic Editor

PLOS ONE